# SE(3)-Transformers: 3D Roto-Translation Equivariant Attention Networks

**Fabian B. Fuchs**[*†]
Bosch Center for Artificial Intelligence
A2I Lab, Oxford University
`fabian@robots.ox.ac.uk`

**Daniel E. Worrall**[*]
Amsterdam Machine Learning Lab, Philips Lab
University of Amsterdam
`d.e.worrall@uva.nl`

**Volker Fischer**
Bosch Center for Artificial Intelligence
`volker.fischer@de.bosch.com`

**Max Welling**
Amsterdam Machine Learning Lab
University of Amsterdam
`m.welling@uva.nl`

## Abstract

We introduce the SE(3)-Transformer, a variant of the self-attention module for 3D point clouds and graphs, which is *equivariant* under continuous 3D roto-translations. Equivariance is important to ensure stable and predictable performance in the presence of nuisance transformations of the data input. A positive corollary of equivariance is increased weight-tying within the model. The SE(3)-Transformer leverages the benefits of self-attention to operate on large point clouds and graphs with varying number of points, while guaranteeing SE(3)-equivariance for robustness. We evaluate our model on a toy $N$-body particle simulation dataset, showcasing the robustness of the predictions under rotations of the input. We further achieve competitive performance on two real-world datasets, ScanObjectNN and QM9. In all cases, our model outperforms a strong, non-equivariant attention baseline and an equivariant model without attention.

## 1 Introduction

Self-attention mechanisms [31] have enjoyed a sharp rise in popularity in recent years. Their relative implementational simplicity coupled with high efficacy on a wide range of tasks such as language modeling [31], image recognition [18], or graph-based problems [32], make them an attractive component to use. However, their generality of application means that for specific tasks, knowledge of existing underlying structure is unused. In this paper, we propose the *SE(3)-Transformer* shown in Fig. 1, a self-attention mechanism specifically for 3D point cloud and graph data, which adheres to *equivariance constraints*, improving robustness to nuisance transformations and general performance.

Point cloud data is ubiquitous across many fields, presenting itself in diverse forms such as 3D object scans [29], 3D molecular structures [21], or $N$-body particle simulations [14]. Finding neural structures which can adapt to the varying number of points in an input, while respecting the irregular sampling of point positions, is challenging. Furthermore, an important property is that these structures should be invariant to global changes in overall input pose; that is, 3D translations and rotations of the input point cloud should not affect the output. In this paper, we find that the explicit imposition of equivariance constraints on the self-attention mechanism addresses these challenges. The SE(3)-Transformer uses the self-attention mechanism as a data-dependent filter particularly suited for sparse, non-voxelised point cloud data, while respecting and leveraging the symmetries of the task at hand.

---

[*]equal contribution
[†]work done while at the Bosch Center for Artificial Intelligence

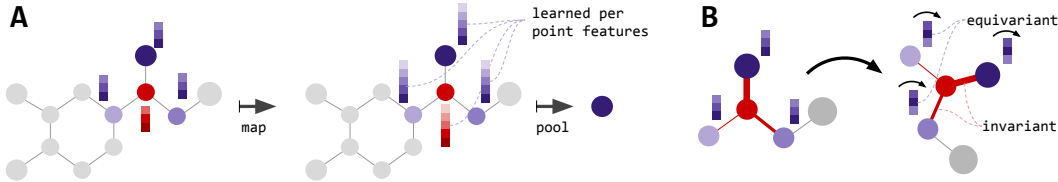

Figure 1: **A**) Each layer of the SE(3)-Transformer maps from a point cloud to a point cloud (or graph to graph) while guaranteeing equivariance. For classification, this is followed by an invariant pooling layer and an MLP. **B**) In each layer, for each node, attention is performed. Here, the red node attends to its neighbours. Attention weights (indicated by line thickness) are invariant w.r.t. input rotation.

Self-attention itself is a pseudo-linear map between sets of points. It can be seen to consist of two components: input-dependent *attention weights* and an embedding of the input, called a *value embedding*. In Fig. 1, we show an example of a molecular graph, where attached to every atom we see a value embedding vector and where the attention weights are represented as edges, with width corresponding to the attention weight magnitude. In the SE(3)-Transformer, we explicitly design the attention weights to be invariant to global pose. Furthermore, we design the value embedding to be equivariant to global pose. Equivariance generalises the translational weight-tying of convolutions. It ensures that transformations of a layer's input manifest as equivalent transformations of the output. SE(3)-equivariance in particular is the generalisation of translational weight-tying in 2D known from conventional convolutions to roto-translations in 3D. This restricts the space of learnable functions to a subspace which adheres to the symmetries of the task and thus reduces the number of learnable parameters. Meanwhile, it provides us with a richer form of invariance, since relative positional information between features in the input is preserved.

The works closest related to ours are tensor field networks (TFN) [28] and their voxelised equivalent, 3D steerable CNNs [37]. These provide frameworks for building SE(3)-equivariant convolutional networks operating on point clouds. Employing self-attention instead of convolutions has several advantages. (1) It allows a natural handling of edge features extending TFNs to the graph setting. (2) This is one of the first examples of a nonlinear equivariant layer. In Section 3.2, we show our proposed approach relieves the strong angular constraints on the filter compared to TFNs, therefore adding representational capacity. This constraint has been pointed out in the equivariance literature to limit performance severely [36]. Furthermore, we provide a more efficient implementation, mainly due to a GPU accelerated version of the spherical harmonics. The TFN baselines in our experiments leverage this and use significantly scaled up architectures compared to the ones used in [28].

Our contributions are the following:

- We introduce a novel self-attention mechanism, guaranteeably invariant to global rotations and translations of its input. It is also equivariant to permutations of the input point labels.

- We show that the SE(3)-Transformer resolves an issue with concurrent SE(3)-equivariant neural networks, which suffer from angularly constrained filters.

- We introduce a `Pytorch` implementation of spherical harmonics, which is 10x faster than `Scipy` on CPU and $100 - 1000\times$ faster on GPU. This directly addresses a bottleneck of TFNs [28]. E.g., for a ScanObjectNN model, we achieve $\approx 22\times$ speed up of the forward pass compared to a network built with SH from the `lielearn` library (see Appendix C).

- Code available at `https://github.com/FabianFuchsML/se3-transformer-public`

## 2 Background And Related Work

In this section we introduce the relevant background materials on self-attention, graph neural networks, and equivariance. We are concerned with point cloud based machine learning tasks, such as object classification or segmentation. In such a task, we are given a point cloud as input, represented as a collection of $n$ coordinate vectors $\mathbf{x}_i \in \mathbb{R}^3$ with optional per-point features $\mathbf{f}_i \in \mathbb{R}^d$.

### 2.1 The Attention Mechanism

The standard *attention mechanism* [31] can be thought of as consisting of three terms: a set of query vectors $\mathbf{q}_i \in \mathbb{R}^p$ for $i = 1, ..., m$, a set of key vectors $\mathbf{k}_j \in \mathbb{R}^p$ for $j = 1, ..., n$, and a set of

value vectors $\mathbf{v}_j \in \mathbb{R}^r$ for $j = 1, ..., n$, where $r$ and $p$ are the dimensions of the low dimensional embeddings. We commonly interpret the key $\mathbf{k}_j$ and the value $\mathbf{v}_j$ as being 'attached' to the same point $j$. For a given query $\mathbf{q}_i$, the attention mechanism can be written as

$$\text{Attn}\left(\mathbf{q}_i, \{\mathbf{k}_j\}, \{\mathbf{v}_j\}\right) = \sum_{j=1}^n \alpha_{ij} \mathbf{v}_j, \qquad \alpha_{ij} = \frac{\exp(\mathbf{q}_i^\top \mathbf{k}_j)}{\sum_{j'=1}^n \exp(\mathbf{q}_i^\top \mathbf{k}_{j'})} \qquad (1)$$

where we used a softmax as a nonlinearity acting on the weights. In general, the number of query vectors does not have to equal the number of input points [16]. In the case of *self-attention* the query, key, and value vectors are embeddings of the input features, so

$$\mathbf{q} = h_Q(\mathbf{f}), \qquad\qquad \mathbf{k} = h_K(\mathbf{f}), \qquad\qquad \mathbf{v} = h_V(\mathbf{f}), \qquad (2)$$

where $\{h_Q, h_K, h_V\}$ are, in the most general case, neural networks [30]. For us, query $\mathbf{q}_i$ is associated with a point $i$ in the input, which has a geometric location $\mathbf{x}_i$. Thus if we have $n$ points, we have $n$ possible queries. For query $\mathbf{q}_i$, we say that node $i$ *attends* to all other nodes $j \neq i$.

Motivated by a successes across a wide range of tasks in deep learning such as language modeling [31], image recognition [18], graph-based problems [32], and relational reasoning [30, 9], a recent stream of work has applied forms of self-attention algorithms to point cloud data [44, 42, 16]. One such example is the Set Transformer [16]. When applied to object classification on ModelNet40 [41], the input to the Set Transformer are the cartesian coordinates of the points. Each layer embeds this positional information further while dynamically querying information from other points. The final per-point embeddings are downsampled and used for object classification.

**Permutation equivariance** A key property of self-attention is *permutation equivariance*. Permutations of point labels $1, ..., n$ lead to permutations of the self-attention output. This guarantees the attention output does not depend arbitrarily on input point ordering. Wagstaff et al. [33] recently showed that this mechanism can theoretically approximate *all* permutation equivariant functions. The SE(3)-transformer is a special case of this attention mechanism, inheriting permutation equivariance. However, it limits the space of learnable functions to rotation and translation equivariant ones.

## 2.2 Graph Neural Networks

Attention scales quadratically with point cloud size, so it is useful to introduce neighbourhoods: instead of each point attending to *all* other points, it only attends to its nearest neighbours. Sets with neighbourhoods are naturally represented as graphs. Attention has previously been introduced on graphs under the names of intra-, self-, vertex-, or graph-attention [17, 31, 32, 12, 26]. These methods were unified by Wang et al. [34] with the non-local neural network. This has the simple form

$$\mathbf{y}_i = \frac{1}{\mathcal{C}(\{\mathbf{f}_j \in \mathcal{N}_i\})} \sum_{j \in \mathcal{N}_i} w(\mathbf{f}_i, \mathbf{f}_j) h(\mathbf{f}_j) \qquad (3)$$

where $w$ and $h$ are neural networks and $\mathcal{C}$ normalises the sum as a function of all features in the neighbourhood $\mathcal{N}_i$. This has a similar structure to attention, and indeed we can see it as performing attention per neighbourhood. While non-local modules do not explicitly incorporate edge-features, it is possible to add them, as done in Veličković et al. [32] and Hoshen [12].

## 2.3 Equivariance

Given a set of transformations $T_g : \mathcal{V} \to \mathcal{V}$ for $g \in G$, where $G$ is an abstract group, a function $\phi : \mathcal{V} \to \mathcal{Y}$ is called equivariant if for every $g$ there exists a transformation $S_g : \mathcal{Y} \to \mathcal{Y}$ such that

$$S_g[\phi(v)] = \phi(T_g[v]) \qquad \text{for all } g \in G, v \in \mathcal{V}. \qquad (4)$$

The indices $g$ can be considered as parameters describing the transformation. Given a pair $(T_g, S_g)$, we can solve for the family of equivariant functions $\phi$ satisfying Equation 4. Furthermore, if $(T_g, S_g)$ are linear and the map $\phi$ is also linear, then a very rich and developed theory already exists for finding $\phi$ [6]. In the equivariance literature, deep networks are built from interleaved linear maps $\phi$ and equivariant nonlinearities. In the case of 3D roto-translations it has already been shown that a suitable structure for $\phi$ is a *tensor field network* [28], explained below. Note that Romero et al. [24] recently introduced a 2D roto-translationally equivariant attention module for pixel-based image data.

**Group Representations** In general, the transformations $(T_g, S_g)$ are called *group representations*. Formally, a group representation $\rho : G \to GL(N)$ is a map from a group $G$ to the set of $N \times N$

invertible matrices $GL(N)$. Critically $\rho$ is a *group homomorphism*; that is, it satisfies the following property $\rho(g_1 g_2) = \rho(g_1)\rho(g_2)$ for all $g_1, g_2 \in G$. Specifically for 3D rotations $G = SO(3)$, we have a few interesting properties: 1) its representations are orthogonal matrices, 2) all representations can be decomposed as

$$\rho(g) = \mathbf{Q}^\top \left[ \bigoplus_\ell \mathbf{D}_\ell(g) \right] \mathbf{Q}, \tag{5}$$

where $\mathbf{Q}$ is an orthogonal, $N \times N$, change-of-basis matrix [5]; each $\mathbf{D}_\ell$ for $\ell = 0, 1, 2, ...$ is a $(2\ell+1) \times (2\ell+1)$ matrix known as a Wigner-D matrix[3]; and the $\bigoplus$ is the *direct sum* or concatenation of matrices along the diagonal. The Wigner-D matrices are *irreducible representations* of SO(3)— think of them as the 'smallest' representations possible. Vectors transforming according to $\mathbf{D}_\ell$ (i.e. we set $\mathbf{Q} = \mathbf{I}$), are called *type-$\ell$* vectors. Type-0 vectors are invariant under rotations and type-1 vectors rotate according to 3D rotation matrices. Note, type-$\ell$ vectors have length $2\ell + 1$. They can be stacked, forming a feature vector $\mathbf{f}$ transforming according to Eq. (5).

**Tensor Field Networks** Tensor field networks (TFN) [28] are neural networks, which map point clouds to point clouds under the constraint of SE(3)-equivariance, the group of 3D rotations and translations. For point clouds, the input is a vector field $\mathbf{f} : \mathbb{R}^3 \to \mathbb{R}^d$ of the form

$$\mathbf{f}(\mathbf{x}) = \sum_{j=1}^N \mathbf{f}_j \delta(\mathbf{x} - \mathbf{x}_j), \tag{6}$$

where $\delta$ is the Dirac delta function, $\{\mathbf{x}_j\}$ are the 3D point coordinates and $\{\mathbf{f}_j\}$ are point features, representing such quantities as atomic number or point identity. For equivariance to be satisfied, the features of a TFN transform under Eq. (5), where $\mathbf{Q} = \mathbf{I}$. Each $\mathbf{f}_j$ is a concatenation of vectors of different *types*, where a subvector of type-$\ell$ is written $\mathbf{f}_j^\ell$. A TFN layer computes the convolution of a continuous-in-space, learnable weight kernel $\mathbf{W}^{\ell k} : \mathbb{R}^3 \to \mathbb{R}^{(2\ell+1) \times (2k+1)}$ from type-$k$ features to type-$\ell$ features. The type-$\ell$ output of the TFN layer at position $\mathbf{x}_i$ is

$$\mathbf{f}_{\text{out},i}^\ell = \sum_{k \geq 0} \underbrace{\int \mathbf{W}^{\ell k}(\mathbf{x}' - \mathbf{x}_i) \mathbf{f}_{\text{in}}^k(\mathbf{x}') \, d\mathbf{x}'}_{k \to \ell \text{ convolution}} = \sum_{k \geq 0} \sum_{j=1}^n \underbrace{\mathbf{W}^{\ell k}(\mathbf{x}_j - \mathbf{x}_i) \mathbf{f}_{\text{in},j}^k}_{\text{node } j \,\to\, \text{node } i \text{ message}}, \tag{7}$$

We can also include a sum over input channels, but we omit it here. Weiler et al. [37], Thomas et al. [28] and Kondor [15] showed that the kernel $\mathbf{W}^{\ell k}$ lies in the span of an equivariant basis $\{\mathbf{W}_J^{\ell k}\}_{J=|k-\ell|}^{k+\ell}$. The kernel is a linear combination of these basis kernels, where the $J^{\text{th}}$ coefficient is a learnable function $\varphi_J^{\ell k} : \mathbb{R}_{\geq 0} \to \mathbb{R}$ of the radius $\|\mathbf{x}\|$. Mathematically this is

$$\mathbf{W}^{\ell k}(\mathbf{x}) = \sum_{J=|k-\ell|}^{k+\ell} \varphi_J^{\ell k}(\|\mathbf{x}\|) \mathbf{W}_J^{\ell k}(\mathbf{x}), \qquad \text{where } \mathbf{W}_J^{\ell k}(\mathbf{x}) = \sum_{m=-J}^J Y_{Jm}(\mathbf{x}/\|\mathbf{x}\|) \mathbf{Q}_{Jm}^{\ell k}. \tag{8}$$

Each basis kernel $\mathbf{W}_J^{\ell k} : \mathbb{R}^3 \to \mathbb{R}^{(2\ell+1) \times (2k+1)}$ is formed by taking a linear combination of Clebsch-Gordan matrices $\mathbf{Q}_{Jm}^{\ell k}$ of shape $(2\ell + 1) \times (2k + 1)$, where the $J, m^{\text{th}}$ linear combination coefficient is the $m^{\text{th}}$ dimension of the $J^{\text{th}}$ spherical harmonic $Y_J : \mathbb{R}^3 \to \mathbb{R}^{2J+1}$. Each basis kernel $\mathbf{W}_J^{\ell k}$ completely constrains the form of the learned kernel in the angular direction, leaving the only learnable degree of freedom in the radial direction. Note that $\mathbf{W}_J^{\ell k}(\mathbf{0}) \neq \mathbf{0}$ only when $k = \ell$ and $J = 0$, which reduces the kernel to a scalar $w$ multiplied by the identity, $\mathbf{W}^{\ell \ell} = w^{\ell \ell} \mathbf{I}$, referred to as *self-interaction* [28]. As such we can rewrite the TFN layer as

$$\mathbf{f}_{\text{out},i}^\ell = \underbrace{w^{\ell \ell} \mathbf{f}_{\text{in},i}^\ell}_{\text{self-interaction}} + \sum_{k \geq 0} \sum_{j \neq i}^n \mathbf{W}^{\ell k}(\mathbf{x}_j - \mathbf{x}_i) \mathbf{f}_{\text{in},j}^k, \tag{9}$$

Eq. (7) and Eq. (9) present the convolution in message-passing form, where messages are aggregated from all nodes and feature types. They are also a form of nonlocal graph operation as in Eq. (3), where the weights are functions on edges and the features $\{\mathbf{f}_i\}$ are node features. We will later see how our proposed attention layer unifies aspects of convolutions and graph neural networks.

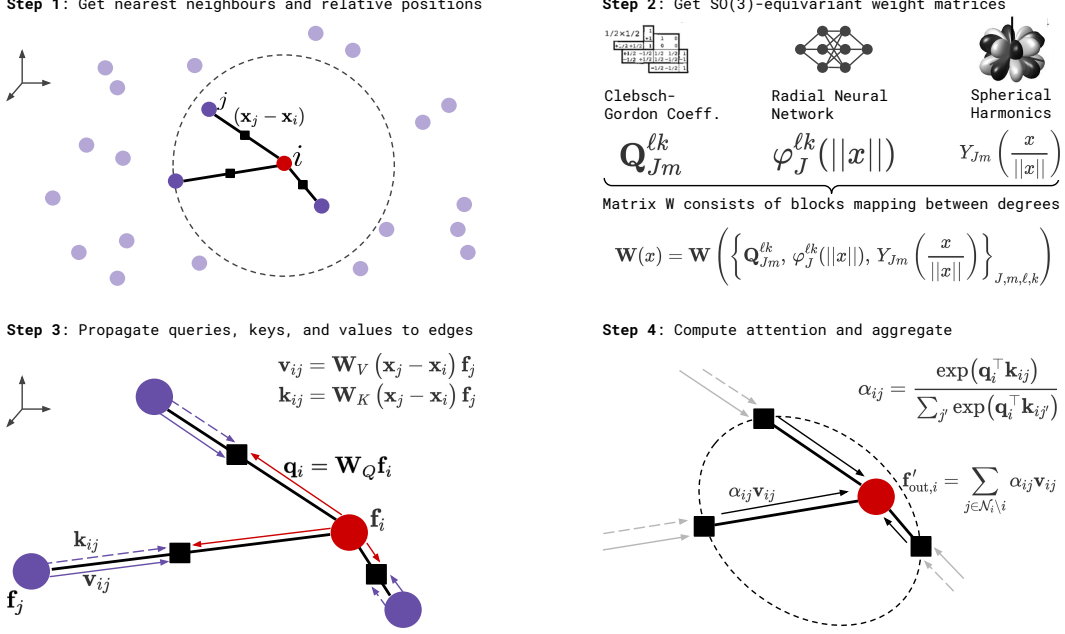

Figure 2: Updating the node features using our equivariant attention mechanism in four steps. A more detailed description, especially of step 2, is provided in the Appendix. Steps 3 and 4 visualise a graph network perspective: features are passed from nodes to edges to compute keys, queries and values, which depend both on features and relative positions in a rotation-equivariant manner.

## 3 Method

Here, we present the *SE(3)-Transformer*. The layer can be broken down into a procedure of steps as shown in Fig. 2, which we describe in the following section. These are the construction of a graph from a point cloud, the construction of equivariant edge functions on the graph, how to propagate SE(3)-equivariant messages on the graph, and how to aggregate them. We also introduce an alternative for the self-interaction layer, which we call *attentive self-interaction*.

### 3.1 Neighbourhoods

Given a point cloud $\{(\mathbf{x}_i, \mathbf{f}_i)\}$, we first introduce a collection of neighbourhoods $\mathcal{N}_i \subseteq \{1, ..., N\}$, one centered on each point $i$. These neighbourhoods are computed either via the nearest-neighbours methods or may already be defined. For instance, molecular structures have neighbourhoods defined by their bonding structure. Neighbourhoods reduce the computational complexity of the attention mechanism from quadratic in the number of points to linear. The introduction of neighbourhoods converts our point cloud into a graph. This step is shown as Step 1 of Fig. 2.

### 3.2 The SE(3)-Transformer

The SE(3)-Transformer itself consists of three components. These are 1) edge-wise attention weights $\alpha_{ij}$, constructed to be SE(3)-invariant on each edge $ij$, 2) edge-wise SE(3)-equivariant value messages, propagating information between nodes, as found in the TFN convolution of Eq. (7), and 3) a linear/attentive self-interaction layer. Attention is performed on a per-neighbourhood basis as follows:

$$\mathbf{f}^{\ell}_{\text{out},i} = \underbrace{\mathbf{W}^{\ell\ell}_V \mathbf{f}^{\ell}_{\text{in},i}}_{\text{③ self-interaction}} + \sum_{k \geq 0} \sum_{j \in \mathcal{N}_i \setminus i} \underbrace{\alpha_{ij}}_{\text{① attention}} \underbrace{\mathbf{W}^{\ell k}_V (\mathbf{x}_j - \mathbf{x}_i) \mathbf{f}^k_{\text{in},j}}_{\text{② value message}} . \tag{10}$$

These components are visualised in Fig. 2. If we remove the attention weights then we have a tensor field convolution, and if we instead remove the dependence of $\mathbf{W}_V$ on $(\mathbf{x}_j - \mathbf{x}_i)$, we have a conventional attention mechanism. Provided that the attention weights $\alpha_{ij}$ are invariant, Eq. (10) is equivariant to SE(3)-transformations. This is because it is just a linear combination of equivariant value messages. Invariant attention weights can be achieved with a dot-product attention structure shown in Eq. (11). This mechanism consists of a normalised inner product between a query vector $\mathbf{q}_i$

at node $i$ and a set of key vectors $\{\mathbf{k}_{ij}\}_{j \in \mathcal{N}_i}$ along each edge $ij$ in the neighbourhood $\mathcal{N}_i$ where

$$\alpha_{ij} = \frac{\exp(\mathbf{q}_i^\top \mathbf{k}_{ij})}{\sum_{j' \in \mathcal{N}_i \setminus i} \exp(\mathbf{q}_i^\top \mathbf{k}_{ij'})}, \quad \mathbf{q}_i = \bigoplus_{\ell \geq 0} \sum_{k \geq 0} \mathbf{W}_Q^{\ell k} \mathbf{f}_{\text{in},i}^k, \quad \mathbf{k}_{ij} = \bigoplus_{\ell \geq 0} \sum_{k \geq 0} \mathbf{W}_K^{\ell k}(\mathbf{x}_j - \mathbf{x}_i) \mathbf{f}_{\text{in},j}^k. \quad (11)$$

$\bigoplus$ is the direct sum, i.e. vector concatenation in this instance. The linear embedding matrices $\mathbf{W}_Q^{\ell k}$ and $\mathbf{W}_K^{\ell k}(\mathbf{x}_j - \mathbf{x}_i)$ are of TFN type (c.f. Eq. (8)). The attention weights $\alpha_{ij}$ are invariant for the following reason. If the input features $\{\mathbf{f}_{\text{in},j}\}$ are SO(3)-equivariant, then the query $\mathbf{q}_i$ and key vectors $\{\mathbf{k}_{ij}\}$ are also SE(3)-equivariant, since the linear embedding matrices are of TFN type. The inner product of SO(3)-equivariant vectors, transforming under the same representation $\mathbf{S}_g$ is invariant, since if $\mathbf{q} \mapsto \mathbf{S}_g \mathbf{q}$ and $\mathbf{k} \mapsto \mathbf{S}_g \mathbf{k}$, then $\mathbf{q}^\top \mathbf{S}_g^\top \mathbf{S}_g \mathbf{k} = \mathbf{q}^\top \mathbf{k}$, because of the orthonormality of representations of SO(3), mentioned in the background section. We follow the common practice from the self-attention literature [31, 16], and chosen a softmax nonlinearity to normalise the attention weights to unity, but in general any nonlinear function could be used.

**Aside: Angular Modulation** The attention weights add extra degrees of freedom to the TFN kernel in the angular direction. This is seen when Eq. (10) is viewed as a convolution with a data-dependent kernel $\alpha_{ij} \mathbf{W}_V^{\ell k}(\mathbf{x})$. In the literature, SO(3) equivariant kernels are decomposed as a sum of products of learnable radial functions $\varphi_J^{\ell k}(\|\mathbf{x}\|)$ and non-learnable angular kernels $\mathbf{W}_J^{\ell k}(\mathbf{x}/\|\mathbf{x}\|)$ (c.f. Eq. (8)). The fixed angular dependence of $\mathbf{W}_J^{\ell k}(\mathbf{x}/\|\mathbf{x}\|)$ is a strange artifact of the equivariance condition in noncommutative algebras and while necessary to guarantee equivariance, it is seen as overconstraining the expressiveness of the kernels. Interestingly, the attention weights $\alpha_{ij}$ introduce a means to modulate the angular profile of $\mathbf{W}_J^{\ell k}(\mathbf{x}/\|\mathbf{x}\|)$, while maintaining equivariance.

**Channels, Self-interaction Layers, and Non-Linearities** Analogous to conventional neural networks, the SE(3)-Transformer can straightforwardly be extended to multiple channels per representation degree $\ell$, so far omitted for brevity. This sets the stage for self-interaction layers. The attention layer (c.f. Fig. 2 and circles 1 and 2 of Eq. (10)) aggregates information over nodes and input representation degrees $k$. In contrast, the self-interaction layer (c.f. circle 3 of Eq. (10)) exchanges information solely between features of the same degree and within one node—much akin to 1x1 convolutions in CNNs. Self-interaction is an elegant form of learnable skip connection, transporting information from query point $i$ in layer $L$ to query point $i$ in layer $L + 1$. This is crucial since, in the SE(3)-Transformer, points do not attend to themselves. In our experiments, we use two different types of self-interaction layer: (1) linear and (2) attentive, both of the form

$$\mathbf{f}_{\text{out},i,c'}^\ell = \sum_c w_{i,c'c}^{\ell\ell} \mathbf{f}_{\text{in},i,c}^\ell. \quad (12)$$

**Linear:** Following Schütt et al. [25], output channels are a learned linear combination of input channels using one set of weights $w_{i,c'c}^{\ell\ell} = w_{c'c}^{\ell\ell}$ per representation degree, shared across all points. As proposed in Thomas et al. [28], this is followed by a norm-based non-linearity.

**Attentive**: We propose an extension of linear self-interaction, *attentive self-interaction*, combining self-interaction and nonlinearity. We replace the learned scalar weights $w_{c'c}^{\ell\ell}$ with attention weights output from an MLP, shown in Eq. (13) ($\bigoplus$ means concatenation.). These weights are SE(3)-invariant due to the invariance of inner products of features, transforming under the same representation.

$$w_{i,c'c}^{\ell\ell} = \text{MLP}\left(\bigoplus_{c,c'} \mathbf{f}_{\text{in},i,c'}^{\ell\top} \mathbf{f}_{\text{in},i,c}^\ell\right) \quad (13)$$

### 3.3 Node and Edge Features

Point cloud data often has information attached to points (node-features) and connections between points (edge-features), which we would both like to pass as inputs into the first layer of the network. Node information can directly be incorporated via the tensors $\mathbf{f}_j$ in Eqs. (6) and (10). For incorporating edge information, note that $\mathbf{f}_j$ is part of multiple neighbourhoods. One can replace $\mathbf{f}_j$ with $\mathbf{f}_{ij}$ in Eq. (10). Now, $\mathbf{f}_{ij}$ can carry different information depending on which neighbourhood $\mathcal{N}_i$ we are currently performing attention over. In other words, $\mathbf{f}_{ij}$ can carry information both about node $j$ but also about edge $ij$. Alternatively, if the edge information is scalar, it can be incorporated into the weight matrices $\mathbf{W}_V$ and $\mathbf{W}_K$ as an input to the radial network (see step 2 in Fig. 2).

Table 1: Predicting future locations and velocities in an electron-proton simulation.

| | | Linear | DeepSet [46] | Tensor Field [28] | Set Transformer [16] | **SE(3)-Transformer** |
|---|---|---|---|---|---|---|
| **Position** | MSE $x$ | 0.0691 | 0.0639 | 0.0151 | 0.0139 | **0.0076** |
| | std | - | 0.0086 | 0.0011 | 0.0004 | 0.0002 |
| | $\Delta_{EQ}$ | - | 0.038 | $1.9 \cdot 10^{-7}$ | 0.167 | $3.2 \cdot 10^{-7}$ |
| **Velocity** | MSE $v$ | 0.261 | 0.246 | 0.125 | 0.101 | **0.075** |
| | std | - | 0.017 | 0.002 | 0.004 | 0.001 |
| | $\Delta_{EQ}$ | - | 1.11 | $5.0 \cdot 10^{-7}$ | 0.37 | $6.3 \cdot 10^{-7}$ |

## 4  Experiments

We test the efficacy of the SE(3)-Transformer on three datasets, each testing different aspects of the model. The N-body problem is an equivariant task: rotation of the input should result in rotated predictions of locations and velocities of the particles. Next, we evaluate on a real-world object classification task. Here, the network is confronted with large point clouds of noisy data with symmetry only around the gravitational axis. Finally, we test the SE(3)-Transformer on a molecular property regression task, which shines light on its ability to incorporate rich graph structures. We compare to publicly available, state-of-the-art results as well as a set of our own baselines. Specifically, we compare to the Set-Transformer [16], a non-equivariant attention model, and Tensor Field Networks [28], which is similar to SE(3)-Transformer but does not leverage attention.

Similar to [27, 39], we measure the exactness of equivariance by applying uniformly sampled SO(3)-transformations to input and output. The distance between the two, averaged over samples, yields the equivariance error. Note that, unlike in Sosnovik et al. [27], the error is not squared:

$$\Delta_{EQ} = \|L_s \Phi(f) - \Phi L_s(f)\|_2 \,/\, \|L_s \Phi(f)\|_2 \tag{14}$$

### 4.1  N-Body Simulations

In this experiment, we use an adaptation of the dataset from Kipf et al. [14]. Five particles each carry either a positive or a negative charge and exert repulsive or attractive forces on each other. The input to the network is the position of a particle in a specific time step, its velocity, and its charge. The task of the algorithm is then to predict the relative location and velocity 500 time steps into the future. We deliberately formulated this as a regression problem to avoid the need to predict multiple time steps iteratively. Even though it certainly is an interesting direction for future research to combine equivariant attention with, e.g., an LSTM, our goal here was to test our core contribution and compare it to related models. This task sets itself apart from the other two experiments by not being invariant but equivariant: When the input is rotated or translated, the output changes respectively (see Fig. 3).

We trained an SE(3)-Transformer with 4 equivariant layers, each followed by an attentive self-interaction layer (details are provided in the Appendix). Table 1 shows quantitative results. Our model outperforms both an attention-based, but not rotation-equivariant approach (Set Transformer) and a equivariant approach which does not leverage attention (Tensor Field). The equivariance error shows that our approach is indeed fully rotation equivariant up to the precision of the computations.

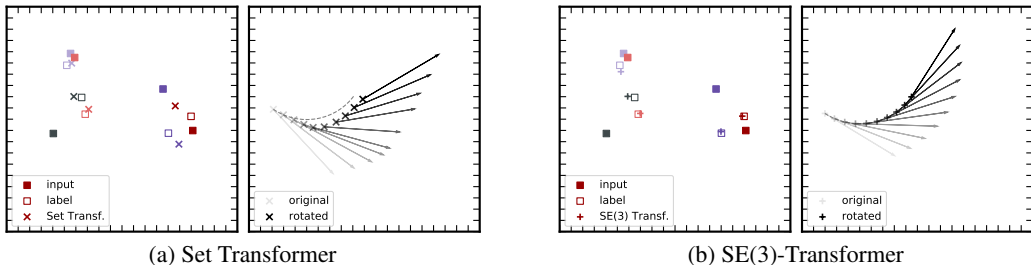

(a) Set Transformer          (b) SE(3)-Transformer

Figure 3: A model based on conventional self-attention (left) and our rotation-equivariant version (right) predict future locations and velocities in a 5-body problem. The respective left-hand plots show input locations at time step $t = 0$, ground truth locations at $t = 500$, and the respective predictions. The right-hand plots show predicted locations and velocities for rotations of the input in steps of 10 degrees. The dashed curves show the predicted locations of a perfectly equivariant model.

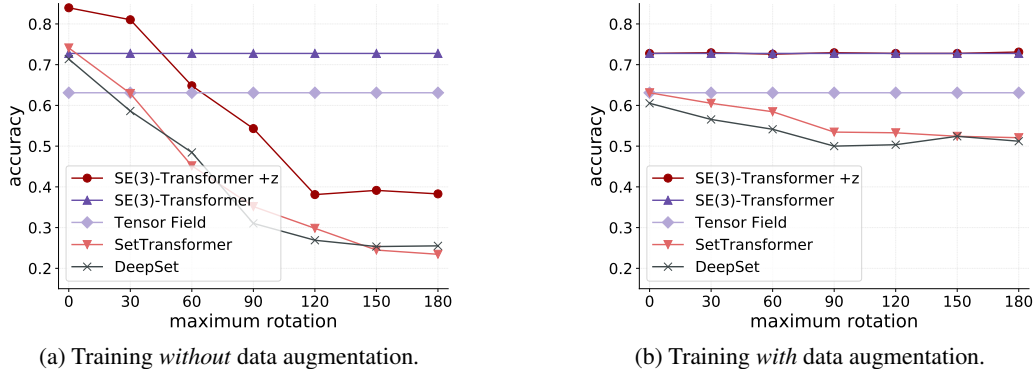

(a) Training *without* data augmentation.  (b) Training *with* data augmentation.

Figure 4: ScanObjectNN: $x$-axis shows data augmentation on the test set. The $x$-value corresponds to the maximum rotation around a random axis in the $x$-$y$-plane. If both training and test set are not rotated ($x = 0$ in **a**), breaking the symmetry of the SE(3)-Transformer by providing the $z$-component of the coordinates as an additional, scalar input improves the performance significantly. Interestingly, the model learns to ignore the additional, symmetry-breaking input when the training set presents a rotation-invariant problem (strongly overlapping dark red circles and dark purple triangles in **b**).

Table 2: Classification accuracy on the 'object only' category of the ScanObjectNN dataset[4]. The performance of the SE(3)-Transformer is averaged over 5 runs (standard deviation 0.7%).

| | Tensor Field | DeepSet | SE(3)-Transf. | 3DmFV | Set Transformer | PointNet | SpiderCNN | Tensor Field +z | PointNet++ | SE(3)-Transf.+z | PointCNN | DGCNN | PointGLR |
|---|---|---|---|---|---|---|---|---|---|---|---|---|---|
| No. Points | 128 | 1024 | **128** | 1024 | 1024 | 1024 | 1024 | 128 | 1024 | **128** | 1024 | 1024 | 1024 |
| Accuracy | 63.1% | 71.4% | **72.8** % | 73.8% | 74.1% | 79.2% | 79.5% | 81.0% | 84.3% | **85.0%** | 85.5% | 86.2% | 87.2% |

## 4.2 Real-World Object Classification on ScanObjectNN

Object classification from point clouds is a well-studied task. Interestingly, the vast majority of current neural network methods work on scalar coordinates without incorporating vector specific inductive biases. Some recent works explore rotation invariant point cloud classification [45, 47, 4, 22]. Our method sets itself apart by using roto-translation equivariant layers acting directly on the point cloud without prior projection onto a sphere [22, 45, 7]. This allows for weight-tying and increased sample efficiency while transporting information about local and global orientation through the network - analogous to the translation equivariance of CNNs on 2D images. To test our method, we choose ScanObjectNN, a recently introduced dataset for real-world object classification. The benchmark provides point clouds of 2902 objects across 15 different categories. We only use the coordinates of the points as input and object categories as training labels. We train an SE(3)-Transformer with 4 equivariant layers with linear self-interaction followed by max-pooling and an MLP. Interestingly, the task is not fully rotation invariant, in a statistical sense, as the objects are aligned with respect to the gravity axis. This results in a performance loss when deploying a fully SO(3) invariant model (see Fig. 4a). In other words: when looking at a new object, it helps to know where 'up' is. We create an SO(2) invariant version of our algorithm by additionally feeding the $z$-component as an type-0 field and the $x$, $y$ position as an additional type-1 field (see Appendix). We dub this model *SE(3)-Transformer +z*. This way, the model can 'learn' which symmetries to adhere to by suppressing and promoting different inputs (compare Fig. 4a and Fig. 4b). In Table 2, we compare our model to the current state-of-the-art in object classification[4]. Despite the dataset not playing to the strengths of our model (full SE(3)-invariance) and a much lower number of input points, the performance is competitive with models specifically designed for object classification - PointGLR [23], for instance, is pre-trained on the larger ModelNet40 dataset [41]. For a discussion of performance vs. number of input points used, see Appendix D.1.2.

## 4.3 QM9

The QM9 regression dataset [21] is a publicly available chemical property prediction task. There are 134k molecules with up to 29 atoms per molecule. Atoms are represented as a 5 dimensional one-hot node embeddings in a molecular graph connected by 4 different chemical bond types (more details in Appendix). 'Po-

Table 3: QM9 Mean Absolute Error. Top: Non-equivariant models. Bottom: Equivariant models. SE(3)-Tr. is averaged over 5 runs.

| TASK<br>UNITS | $\alpha$<br>bohr$^3$ | $\Delta\varepsilon$<br>meV | $\varepsilon_{\text{HOMO}}$<br>meV | $\varepsilon_{\text{LUMO}}$<br>meV | $\mu$<br>D | $C_\nu$<br>cal/mol K |
|---|---|---|---|---|---|---|
| WaveScatt [11] | .160 | 118 | 85 | 76 | .340 | .049 |
| NMP [10] | .092 | 69 | 43 | 38 | .030 | .040 |
| SchNet [25] | .235 | 63 | 41 | 34 | .033 | .033 |
| Cormorant [1] | .085 | 61 | 34 | 38 | .038 | .026 |
| LieConv(T3) [8] | .084 | 49 | 30 | 25 | .032 | .038 |
| TFN [28] | .223 | 58 | 40 | 38 | .064 | .101 |
| **SE(3)-Transformer** | .142±.002 | 53.0±0.3 | 33.0±.9 | 35.0±.7 | .051±.001 | .054±.002 |

sitions' of each atom are provided. We show results on the test set of Anderson et al. [1] for 6 regression tasks in Table 3. Lower is better. The table is split into non-equivariant (top) and equivariant models (bottom). Our nearest models are Cormorant and TFN (own implementation). We see that while not state-of-the-art, we offer competitive performance, especially against Cormorant and TFN, which transform under irreducible representations of SE(3) (like us), unlike LieConv(T3), using a left-regular representation of SE(3), which may explain its success.

## 5 Conclusion

We have presented an attention-based neural architecture designed specifically for point cloud data. This architecture is guaranteed to be robust to rotations and translations of the input, obviating the need for training time data augmentation and ensuring stability to arbitrary choices of coordinate frame. The use of self-attention allows for anisotropic, data-adaptive filters, while the use of neighbourhoods enables scalability to large point clouds. We have also introduced the interpretation of the attention mechanism as a data-dependent nonlinearity, adding to the list of equivariant nonlinearties which we can use in equivariant networks. Furthermore, we provide code for a speed up of spherical harmonics computation of up to 3 orders of magnitudes. This speed-up allowed us to train significantly larger versions of both the SE(3)-Transformer and the Tensor Field network [28] and to apply these models to real-world datasets.

Our experiments showed that adding attention to a roto-translation-equivariant model consistently led to higher accuracy and increased training stability. Specifically for large neighbourhoods, attention proved essential for model convergence. On the other hand, compared to convential attention, adding the equivariance constraints also increases performance in all of our experiments while at the same time providing a mathematical guarantee for robustness with respect to rotations of the input data.

## Broader Impact

The main contribution of the paper is a mathematically motivated attention mechanism which can be used for deep learning on point cloud based problems. We do not see a direct potential of *negative* impact to the society. However, we would like to stress that this type of algorithm is inherently suited for classification and regression problems on molecules. The SE(3)-Transformer therefore lends itself to application in drug research. One concrete application we are currently investigating is to use the algorithm for early-stage suitability classification of molecules for inhibiting the reproductive cycle of the coronavirus. While research of this sort always requires intensive testing in wet labs, computer algorithms can be and are being used to filter out particularly promising compounds from large databases of millions of molecules.

## Acknowledgements and Funding Disclosure

We would like to express our gratitude to the Bosch Center for Artificial Intelligence and Koninclijke Philips N.V. for funding our work and contributing to open research by publishing our paper. Fabian Fuchs worked on this project while on a research sabbatical at the Bosch Center for Artificial Intelligence. His PhD is funded by the EPSRC AIMS Centre for Doctoral Training at Oxford University.

## Footnotes

[3]The 'D' stands for *Darstellung*, German for representation

[4]At time of submission, PointGLR was a recently published preprint [23]. The performance of the following models was taken from the official benchmark of the dataset as of June 4th, 2020 (`https://hkust-vgd.github.io/benchmark/`): 3DmFV [3], PointNet [19], SpiderCNN [43], PointNet++ [20], DGCN [35].

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
