[Supplementary Material]

# A Group Theory and Tensor Field Networks

**Groups** A *group* is an abstract mathematical concept. Formally a group $(G, \circ)$ consists of a set $G$ and a binary composition operator $\circ : G \times G \to G$ (typically we just use the symbol $G$ to refer to the group). All groups must adhere to the following 4 axioms

- **Closure**: $g \circ h \in G$ for all $g, h \in G$
- **Associativity**: $f \circ (g \circ h) = (f \circ g) \circ h = f \circ g \circ h$ for all $f, g, h \in G$
- **Identity**: There exists an element $e \in G$ such that $e \circ g = g \circ e = g$ for all $g \in G$
- **Inverses**: For each $g \in G$ there exists a $g^{-1} \in G$ such that $g^{-1} \circ g = g \circ g^{-1} = e$

In practice, we omit writing the binary composition operator $\circ$, so would write $gh$ instead of $g \circ h$. Groups can be finite or infinite, countable or uncountable, compact or non-compact. Note that they are not necessarily *commutative*; that is, $gh \neq hg$ in general.

**Actions/Transformations** Groups are useful concepts, because they allow us to describe the structure of *transformations*, also sometimes called *actions*. A transformation (operator) $T_g : \mathcal{X} \to \mathcal{X}$ is an injective map from a space into itself. It is parameterised by an element $g$ of a group $G$. Transformations obey two laws:

- **Closure**: $T_g \circ T_h$ is a valid transformation for all $g, h \in G$
- **Identity**: There exists at least one element $e \in G$ such that $T_e[\mathbf{x}] = \mathbf{x}$ for all $\mathbf{x} \in \mathcal{X}$,

where $\circ$ denotes composition of transformations. For the expression $T_g[\mathbf{x}]$, we say that $T_g$ *acts* on $\mathbf{x}$. It can also be shown that transformations are associative under composition. To codify the structure of a transformation, we note that due to closure we can always write

$$T_g \circ T_h = T_{gh}, \tag{15}$$

If for any $x, y \in \mathcal{X}$ we can always find a group element $g$, such that $T_g[x] = y$, then we call $\mathcal{X}$ a *homogeneous space*. Homogeneous spaces are important concepts, because to each pair of points $x, y$ we can always associate at least one group element.

**Equivariance and Intertwiners** As written in the main body of the text, equivariance is a property of functions $f : \mathcal{X} \to \mathcal{Y}$. Just to recap, given a set of transformations $T_g : \mathcal{X} \to \mathcal{X}$ for $g \in G$, where $G$ is an abstract group, a function $f : \mathcal{X} \to \mathcal{Y}$ is called equivariant if for every $g$ there exists a transformation $S_g : \mathcal{Y} \to \mathcal{Y}$ such that

$$S_g[f(\mathbf{x})] = f(T_g[\mathbf{x}]) \qquad \text{for all } g \in G, \mathbf{x} \in \mathcal{X}. \tag{16}$$

If $f$ is linear and equivariant, then it is called an intertwiner. Two important questions arise: 1) How do we choose $S_g$? 2) once we have $(T_g, S_g)$, how do we solve for $f$? To answer these questions, we need to understand what kinds of $S_g$ are possible. For this, we review representations.

**Representations** A group representation $\rho : G \to GL(N)$ is a map from a group $G$ to the set of $N \times N$ invertible matrices $GL(N)$. Critically $\rho$ is a *group homomorphism*; that is, it satisfies the following property $\rho(g_1 g_2) = \rho(g_1)\rho(g_2)$ for all $g_1, g_2 \in G$. Representations can be used as transformation operators, acting on $N$-dimensional vectors $\mathbf{x} \in \mathbb{R}^N$. For instance, for the group of 3D rotations, known as $SO(3)$, we have that 3D rotation matrices, $\rho(g) = \mathbf{R}_g$ act on (i.e., rotate) 3D vectors, as

$$T_g[\mathbf{x}] = \rho(g)\mathbf{x} = \mathbf{R}_g \mathbf{x}, \qquad \text{for all } \mathbf{x} \in \mathcal{X}, g \in G. \tag{17}$$

However, there are many more representations of $SO(3)$ than just the 3D rotation matrices. Among representations, two representations $\rho$ and $\rho'$ of the same dimensionality are said to be *equivalent* if they can be connected by a similarity transformation

$$\rho'(g) = \mathbf{Q}^{-1}\rho(g)\mathbf{Q}, \qquad \text{for all } g \in G. \tag{18}$$

We also say that a representation is *reducible* if is can be written as

$$\rho(g) = \mathbf{Q}^{-1}(\rho_1(g) \oplus \rho_2(g))\mathbf{Q} = \mathbf{Q}^{-1} \begin{bmatrix} \rho_1(g) & \\ & \rho_2(g) \end{bmatrix} \mathbf{Q}, \qquad \text{for all } g \in G. \tag{19}$$

If the representations $\rho_1$ and $\rho_2$ are not reducible, then they are called *irreducible representations* of $G$, or *irreps*. In a sense, they are the atoms among representations, out of which all other representations can be constructed. Note that each irrep acts on a separate subspace, mapping vectors from that space back into it. We say that subspace $\mathcal{X}_\ell \in \mathcal{X}$ is *invariant under* irrep $\rho_\ell$, if $\{\rho_\ell(g)\mathbf{x} \mid \mathbf{x} \in \mathcal{X}_\ell, g \in G\} \subseteq \mathcal{X}_\ell$.

**Representation theory of** $SO(3)$    As it turns out, all linear representations of compact groups[5] (such as $SO(3)$) can be decomposed into a direct sum of irreps, as

$$\rho(g) = \mathbf{Q}^\top \left[ \bigoplus_J \mathbf{D}_J(g) \right] \mathbf{Q}, \tag{20}$$

where $\mathbf{Q}$ is an orthogonal, $N \times N$, change-of-basis matrix [5]; and each $\mathbf{D}_J$ for $J = 0, 1, 2, ...$ is a $(2J + 1) \times (2J + 1)$ matrix known as a Wigner-D matrix. The Wigner-D matrices are the irreducible representations of $SO(3)$. We also mentioned that vectors transforming according to $\mathbf{D}_J$ (i.e. we set $\mathbf{Q} = \mathbf{I}$), are called *type-J* vectors. Type-0 vectors are invariant under rotations and type-1 vectors rotate according to 3D rotation matrices. Note, type-$J$ vectors have length $2J + 1$. In the previous paragraph we mentioned that irreps act on orthogonal subspaces $\mathcal{X}_0, \mathcal{X}_1, ....$. The orthogonal subspaces corresponding to the Wigner-D matrices are the space of *spherical harmonics*.

**The Spherical Harmonics**    The spherical harmonics $\mathbf{Y}_J : S^2 \to \mathbb{C}^{2J+1}$ for $J \geq 0$ are square-integrable complex-valued functions on the sphere $S^2$. They have the satisfying property that they are rotated directly by the Wigner-D matrices as

$$\mathbf{Y}_J(\mathbf{R}_g^{-1}\mathbf{x}) = \mathbf{D}_J^*(g)\mathbf{Y}_J(\mathbf{x}), \qquad \mathbf{x} \in S^2, g \in G, \tag{21}$$

where $\mathbf{D}_J$ is the $J^\text{th}$ Wigner-D matrix and $\mathbf{D}_J^*$ is its complex conjugate. They form an orthonormal basis for (the Hilbert space of) square-integrable functions on the sphere $L^2(S^2)$, with inner product given as

$$\langle f, h \rangle_{S^2} = \int_{S^2} f(\mathbf{x})h^*(\mathbf{x})\,\mathrm{d}\mathbf{x}. \tag{22}$$

So $\langle Y_{Jm}, Y_{J'm'} \rangle_{S^2} = \delta_{JJ'}\delta_{mm'}$, where $Y_{Jm}$ is the $m^\text{th}$ element of $\mathbf{Y}_J$. We can express any function in $L^2(S^2)$ as a linear combination of spherical harmonics, where

$$f(\mathbf{x}) = \sum_{J \geq 0} \mathbf{f}_J^\top \mathbf{Y}_J(\mathbf{x}), \qquad \mathbf{x} \in S^2, \tag{23}$$

where each $\mathbf{f}_J$ is a vector of coefficients of length $2J + 1$. And in the opposite direction, we can retrieve the coefficients as

$$\mathbf{f}_J = \int_{S^2} f(\mathbf{x})\mathbf{Y}_J^*(\mathbf{x})\,\mathrm{d}\mathbf{x} \tag{24}$$

following from the orthonormality of the spherical harmonics. This is in fact a Fourier transform on the sphere and the the vectors $\mathbf{f}_J$ can be considered Fourier coefficients. Critically, we can represent rotated functions as

$$f(\mathbf{R}_g^{-1}\mathbf{x}) = \sum_{J \geq 0} \mathbf{f}_J^\top \mathbf{D}_J^*(g)\mathbf{Y}_J(\mathbf{x}), \qquad \mathbf{x} \in S^2, g \in G. \tag{25}$$

**The Clebsch-Gordan Decomposition**    In the main text we introduced the *Clebsch-Gordan coefficients*. These are used in the construction of the equivariant kernels. They arise in the situation where we have a tensor product of Wigner-D matrices, which as we will see is part of the equivariance constraint on the form of the equivariant kernels. In representation theory a tensor product of representations is also a representation, but since it is not an easy object to work with, we seek to decompose it into a direct sum of irreps, which are easier. This decomposition is of the form of Eq. (20), written

$$\mathbf{D}_k(g) \otimes \mathbf{D}_\ell(g) = \mathbf{Q}^{\ell k \top} \left[ \bigoplus_{J=|k-\ell|}^{k+\ell} \mathbf{D}_J(g) \right] \mathbf{Q}^{\ell k}. \tag{26}$$

In this specific instance, the change of basis matrices $\mathbf{Q}^{\ell k}$ are given the special name of the Clebsch-Gordan coefficients. These can be found in many mathematical physics libraries.

**Tensor Field Layers** In Tensor Field Networks [28] and 3D Steerable CNNs [37], the authors solve for the intertwiners between SO(3) equivariant point clouds. Here we run through the derivation again in our own notation.

We begin with a point cloud $f(\mathbf{x}) = \sum_{j=1}^{N} \mathbf{f}_j \delta(\mathbf{x} - \mathbf{x}_j)$, where $\mathbf{f}_j$ is an equivariant point feature. Let's say that $\mathbf{f}_j$ is a type-$k$ feature, which we write as $\mathbf{f}_j^k$ to remind ourselves of the fact. Now say we perform a convolution $*$ with kernel $\mathbf{W}^{\ell k} : \mathbf{R}^3 \to \mathbb{R}^{(2\ell+1)\times(2k+1)}$, which maps from type-$k$ features to type-$\ell$ features. Then

$$\mathbf{f}_{\text{out},i}^{\ell} = [\mathbf{W}^{\ell k} * \mathbf{f}_{\text{in}}^k](\mathbf{x}) \tag{27}$$

$$= \int_{\mathbb{R}^3} \mathbf{W}^{\ell k}(\mathbf{x}' - \mathbf{x}_i)\mathbf{f}_{\text{in}}^k(\mathbf{x}') \, d\mathbf{x}' \tag{28}$$

$$= \int_{\mathbb{R}^3} \mathbf{W}^{\ell k}(\mathbf{x}' - \mathbf{x}_i) \sum_{j=1}^{N} \mathbf{f}_{\text{in},j}^k \delta(\mathbf{x}' - \mathbf{x}_j) \, d\mathbf{x}' \tag{29}$$

$$= \sum_{j=1}^{N} \int_{\mathbb{R}^3} \mathbf{W}^{\ell k}(\mathbf{x}' - \mathbf{x}_i)\mathbf{f}_{\text{in},j}^k \delta(\mathbf{x}' - \mathbf{x}_j) \, d\mathbf{x}' \qquad \text{change of variables } \mathbf{x}'' = \mathbf{x}' - \mathbf{x}_j \tag{30}$$

$$= \sum_{j=1}^{N} \int_{\mathbb{R}^3} \mathbf{W}^{\ell k}(\mathbf{x}'' + \mathbf{x}_j - \mathbf{x}_i)\mathbf{f}_{\text{in},j}^k \delta(\mathbf{x}'') \, d\mathbf{x}'' \qquad \text{sifting theorem} \tag{31}$$

$$= \sum_{j=1}^{N} \mathbf{W}^{\ell k}(\mathbf{x}_j - \mathbf{x}_i)\mathbf{f}_{\text{in},j}^k. \tag{32}$$

Now let's apply the equivariance condition to this expression, then

$$\mathbf{D}_\ell(g)\mathbf{f}_{\text{out},i}^{\ell} = \sum_{j=1}^{N} \mathbf{W}^{\ell k}(\mathbf{R}_g^{-1}(\mathbf{x}_j - \mathbf{x}_i))\mathbf{D}_k(g)\mathbf{f}_{\text{in},j}^k \tag{33}$$

$$\implies \mathbf{f}_{\text{out},i}^{\ell} = \sum_{j=1}^{N} \mathbf{D}_\ell(g)^{-1}\mathbf{W}^{\ell k}(\mathbf{R}_g^{-1}(\mathbf{x}_j - \mathbf{x}_i))\mathbf{D}_k(g)\mathbf{f}_{\text{in},j}^k \tag{34}$$

Now we notice that this expression should also be equal to Eq. (32), which is the convolution with an unrotated point cloud. Thus we end up at

$$\boxed{\mathbf{W}^{\ell k}(\mathbf{R}_g^{-1}\mathbf{x}) = \mathbf{D}_\ell(g)\mathbf{W}^{\ell k}(\mathbf{x})\mathbf{D}_k(g)^{-1},} \tag{35}$$

which is sometimes refered to as the *kernel constraint*. To solve the kernel constraint, we notice that it is a linear equation and that we can rearrange it as

$$\text{vec}(\mathbf{W}^{\ell k}(\mathbf{R}_g^{-1}\mathbf{x})) = (\mathbf{D}_k(g) \otimes \mathbf{D}_\ell(g))\text{vec}(\mathbf{W}^{\ell k}(\mathbf{x})) \tag{36}$$

where we used the identity $\text{vec}(\mathbf{AXB}) = (\mathbf{B}^\top \otimes \mathbf{A})\text{vec}(\mathbf{X})$ and the fact that the Wigner-D matrices are orthogonal. Using the Clebsch-Gordan decomposition we rewrite this as

$$\text{vec}(\mathbf{W}^{\ell k}(\mathbf{R}_g^{-1}\mathbf{x})) = \mathbf{Q}^{\ell k\top} \left[ \bigoplus_{J=|k-\ell|}^{k+\ell} \mathbf{D}_J(g) \right] \mathbf{Q}^{\ell k}\text{vec}(\mathbf{W}^{\ell k}(\mathbf{R}_g^{-1}\mathbf{x})). \tag{37}$$

Lastly, we can left multiply both sides by $\mathbf{Q}^{\ell k}$ and denote $\boldsymbol{\eta}^{\ell k}(\mathbf{x}) \triangleq \mathbf{Q}^{\ell k}\text{vec}(\mathbf{W}^{\ell k}(\mathbf{x}))$, noting the the Clebsch-Gordan matrices are orthogonal. At the same time we

$$\boldsymbol{\eta}^{\ell k}(\mathbf{R}_g^{-1}\mathbf{x}) = \left[ \bigoplus_{J=|k-\ell|}^{k+\ell} \mathbf{D}_J(g) \right] \boldsymbol{\eta}^{\ell k}(\mathbf{x}). \tag{38}$$

Thus we have that $\boldsymbol{\eta}_J^{\ell k}(\mathbf{R}_g^{-1}\mathbf{x})$ the $J^{\text{th}}$ subvector of $\boldsymbol{\eta}^{\ell k}(\mathbf{R}_g^{-1}\mathbf{x})$ is subject to the constraint

$$\boldsymbol{\eta}_J^{\ell k}(\mathbf{R}_g^{-1}\mathbf{x}) = \mathbf{D}_J(g)\boldsymbol{\eta}_J^{\ell k}(\mathbf{x}), \tag{39}$$

which is exactly the transformation law for the spherical harmonics from Eq. (21)! Thus one way how $\mathbf{W}^{\ell k}(\mathbf{x})$ can be constructed is

$$\text{vec}\left(\mathbf{W}^{\ell k}(\mathbf{x})\right) = \mathbf{Q}^{\ell k\top} \bigoplus_{J=|k-\ell|}^{k+\ell} \mathbf{Y}_J(\mathbf{x}). \tag{40}$$

## B  Recipe for Building an Equivariant Weight Matrix

One of the core operations in the SE(3)-Transformer is multiplying a feature vector $\mathbf{f}$, which transforms according to $SO(3)$, with a matrix $\mathbf{W}$ while preserving equivariance:

$$S_g[\mathbf{W} * \mathbf{f}](\mathbf{x}) = [\mathbf{W} * T_g[\mathbf{f}]](\mathbf{x}), \tag{41}$$

where $T_g[\mathbf{f}](\mathbf{x}) = \rho_{\text{in}}(g)\mathbf{f}(\mathbf{R}_g^{-1}\mathbf{x})$ and $S_g[\mathbf{f}](\mathbf{x}) = \rho_{\text{out}}(g)\mathbf{f}(\mathbf{R}_g^{-1}\mathbf{x})$. Here, as in the previous section we showed how such a matrix $\mathbf{W}$ could be constructed when mapping between features of type-$k$ and type-$\ell$, where $\rho_{\text{in}}(g)$ is a block diagonal matrix of type-$k$ Wigner-D matrices and similarly $\rho_{\text{in}}(g)$ is made of type-$\ell$ Wigner-D matrices. $\mathbf{W}$ is dependent on the relative position $\mathbf{x}$ and underlies the linear equivariance constraints, but is also has learnable components, which we did not show in the previous section. In this section, we show how such a matrix is constructed in practice.

Previously we showed that

$$\text{vec}\left(\mathbf{W}^{\ell k}(\mathbf{x})\right) = \mathbf{Q}^{\ell k\top} \bigoplus_{J=|k-\ell|}^{k+\ell} \mathbf{Y}_J(\mathbf{x}), \tag{42}$$

which is an equivariant mapping between vectors of types $k$ and $\ell$. In practice, we have multiple input vectors $\{\mathbf{f}_c^k\}_c$ of type-$k$ and multiple output vectors of type-$\ell$. For simplicity, however, we ignore this and pretend we only have a single input and single output. Note that $\mathbf{W}^{\ell k}$ has no learnable components. Note that the kernel constraint only acts in the angular direction, but not in the radial direction, so we can introduce scalar radial functions $\varphi_J^{\ell k} : \mathbf{R}_{\geq 0} \to \mathbf{R}$ (one for each $J$), such that

$$\text{vec}\left(\mathbf{W}^{\ell k}(\mathbf{x})\right) = \mathbf{Q}^{\ell k\top} \bigoplus_{J=|k-\ell|}^{k+\ell} \varphi_J^{\ell k}(\|\mathbf{x}\|)\mathbf{Y}_J(\mathbf{x}), \tag{43}$$

The radial functions $\varphi_J^{\ell k}(\|\mathbf{x}\|)$ act as an independent, learnable scalar factor for each degree $J$. The vectorised matrix has dimensionality $(2\ell+1)(2k+1)$. We can unvectorise the above yielding

$$\mathbf{W}^{\ell k}(\mathbf{x}) = \text{unvec}\left(\mathbf{Q}^{\ell k\top} \bigoplus_{J=|k-\ell|}^{k+\ell} \varphi_J^{\ell k}(\|\mathbf{x}\|)\mathbf{Y}_J(\mathbf{x})\right) \tag{44}$$

$$= \sum_{J=|k-\ell|}^{k+\ell} \varphi_J^{\ell k}(\|\mathbf{x}\|)\text{unvec}\left(\mathbf{Q}_J^{\ell k\top}\mathbf{Y}_J(\mathbf{x})\right) \tag{45}$$

where $\mathbf{Q}_J^{\ell k}$ is a $(2\ell+1)(2k+1) \times (2J+1)$ slice from $\mathbf{Q}^{\ell k}$, corresponding to spherical harmonic $\mathbf{Y}_J$. As we showed in the main text, we can also rewrite the unvectorised Clebsch-Gordan–spherical harmonic matrix-vector product as

$$\text{unvec}\left(\mathbf{Q}_J^{\ell k\top}\mathbf{Y}_J(\mathbf{x})\right) = \sum_{m=-J}^{J} \mathbf{Q}_{Jm}^{\ell k\top}Y_{Jm}(\mathbf{x}). \tag{46}$$

In contrast to Weiler et al. [37], we do not voxelise space and therefore $\mathbf{x}$ will be different for each pair of points in each point cloud. However, the same $\mathbf{Y}_J(\mathbf{x})$ will be used multiple times in the network and even multiple times in the same layer. Hence, precomputing them at the beginning of each forward pass for the entire network can significantly speed up the computation. The Clebsch-Gordan coefficients do not depend on the relative positions and can therefore be precomputed once and stored on disk. Multiple libraries exist which approximate those coefficients numerically.

(a) Speed comparison on the CPU.   (b) Speed comparison on the GPU.

Figure 5: Spherical harmonics computation of our own implementation compared to the `lie-learn` library. We found that speeding up the computation of spherical harmonics is critical to scale up both Tensor Field Networks [28] and SE(3)-Transformers to solve real-world machine learning tasks.

## C   Accelerated Computation of Spherical Harmonics

The spherical harmonics (SH) typically have to be computed on the fly for point cloud methods based on irreducible computations, a bottleneck of TFNs [28]. Thomas et al. [28] ameliorate this by restricting the maximum type of feature to type-2, trading expressivity for speed. Weiler et al. [37] circumvent this challenge by voxelising the input, allowing them to pre-compute spherical harmonics for fixed relative positions. This is at the cost of detail as well as exact rotation and translation equivariance.

The number of spherical harmonic lookups in a network based on irreducible representations can quickly become large (number of layers × number of points × number of neighbours × number of channels × number of degrees needed). This motivates parallelised computation on the GPU - a feature not supported by common libraries. To that end, we wrote our own spherical harmonics library in `Pytorch`, which can generate spherical harmonics on the GPU. We found this critical to being able to run the SE(3)-Transformer and Tensor Field network baselines in a reasonable time. This library is accurate to within machine precision against the `scipy` counterpart `scipy.special.sph_harm` and is significantly faster. E.g., for a ScanObjectNN model, we achieve $\sim 22\times$ speed up of the forward pass compared to a network built with SH from the `lielearn` library. A speed comparison isolating the computation of the spherical harmonics is shown in Fig. 5. Code is available at `https://github.com/FabianFuchsML/se3-transformer-public`. In the following, we outline our method to generate them.

The tesseral/real spherical harmonics are given as

$$Y_{Jm}(\theta, \phi) = \sqrt{\frac{2J+1}{4\pi} \frac{(J-m)!}{(J+m)!}} P_J^{|m|}(\cos\theta) \cdot \begin{cases} \sin(|m|\phi) & m < 0, \\ 1 & m = 0, \\ \cos(m\phi) & m > 0, \end{cases} \quad (47)$$

where $P_J^{|m|}$ is the associated Legendre polynomial (ALP), $\theta \in [0, 2\pi)$ is azimuth, and $\phi \in [0, \pi]$ is a polar angle. The term $P_J^{|m|}$ is by far the most expensive component to compute and can be computed recursively. To speed up the computation, we use dynamic programming storing intermediate results in a memoization.

We make use of the following recursion relations in the computation of the ALPs:

$$P_J^{|J|}(x) = (-1)^{|J|} \cdot (1 - x^2)^{|J|/2} \cdot (2|J| - 1)!! \qquad\qquad \text{boundary: } J = m \quad (48)$$

$$P_J^{-m}(x) = (-1)^J \frac{(\ell - m)!}{(\ell + m)!} P_J^m(x) \qquad\qquad \text{negate } m \quad (49)$$

$$P_J^{|m|}(x) = \frac{2J-1}{J-|m|} x P_{J-1}^m(x) + \mathbb{I}[J - |m| > 1] \frac{J + |m| - 1}{J - |m|} P_{J-2}^m(x) \quad \text{recurse} \quad (50)$$

where the *semifactorial* is defined as $x!! = x(x-2)(x-4)\cdots$, and $\mathbb{I}$ is the indicator function. These relations are helpful because they define a recursion.

Figure 6: Subproblem graph for the computatin of the associated Legendre polynomials. To compute $P_3^{-1}(x)$, we compute $P_3^1(x)$, for which we need $P_2^1(x)$ and $P_1^1(x)$. We store each intermediate computation, speeding up average computation time by a factor of $\sim 10$ on CPU.

To understand how we recurse, we consider an example. Fig. 6 shows the space of $J$ and $m$. The black vertices represent a particular ALP, for instance, we have highlighted $P_3^{-1}(x)$. When $m < 0$, we can use Eq. (49) to compute $P_3^{-1}(x)$ from $P_3^1(x)$. We can then use Eq. (50) to compute $P_3^1(x)$ from $P_2^1(x)$ and $P_1^1(x)$. $P_2^1(x)$ can also be computed from Eq. (50) and the boundary value $P_1^1(x)$ can be computed directly using Eq. (48). Crucially, all intermediate ALPs are stored for reuse. Say we wanted to compute $P_4^{-1}(x)$, then we could use Eq. (49) to find it from $P_4^{-1}(x)$, which can be recursed from the stored values $P_3^1(x)$ and $P_2^1(x)$, without needing to recurse down to the boundary.

# D   Experimental Details

## D.1   ScanObjectNN

### D.1.1   SE(3)-Transformer and Tensor Field Network

A particularity of object classification from point clouds is the large number of points the algorithm needs to handle. We use up to 200 points out of the available 2024 points per sample and create neighbourhoods with up to 40 nearest neighbours. It is worth pointing out that especially in this setting, adding self-attention (i.e. when comparing the SE(3) Transformer to Tensor Field Networks) significantly increased the stability. As a result, whenever we swapped out the attention mechanism for a convolution to retrieve the Tensor Field network baseline, we had to decrease the model size to obtain stable training. However, we would like to stress that all the Tensor Field networks we trained were significantly bigger than in the original paper [28], mostly enabled by the faster computation of the spherical harmonics.

For the ablation study in Fig. 4, we trained networks with 4 hidden equivariant layers with 5 channels each, and up to representation degree 2. This results in a hidden feature size per point of

$$5 \cdot \sum_{\ell=0}^{2} (2\ell + 1) = 45 \tag{51}$$

We used 200 points of the point cloud and neighbourhood size 40. For the Tensor Field network baseline, in order to achieve stable training, we used a smaller model with 3 instead of 5 channels, 100 input points and neighbourhood size 10, but with representation degrees up to 3.

We used 1 head per attention mechanism yielding one attention weight for each pair of points but across all channels and degrees (for an implementation of multi-head attention, see Appendix D.3). For the query embedding, we used the identity matrix. For the key embedding, we used a square equivariant matrix preserving the number of degrees and channels per degree.

For the quantitative comparison to the start-of-the-art in Table 2, we used 128 input points and neighbourhood size 10 for both the Tensor Field network baseline and the SE(3)-Transformer. We used farthest point sampling with a random starting point to retrieve the 128 points from the overall

point cloud. We used degrees up to 3 and 5 channels per degree, which we again had to reduce to 3 channels for the Tensor Field network to obtain stable training. We used a norm based non-linearity for the Tensor Field network (as in [28]) and no extra non-linearity (beyond the *softmax* in the self-attention algorithm) for the SE(3) Transformer.

For all experiments, the final layer of the equivariant encoder maps to 64 channels of degree 0 representations. This yields a 64-dimensional SE(3) *invariant* representation per point. Next, we pool over the point dimension followed by an MLP with one hidden layer of dimension 64, a ReLU and a 15 dimensional output with a cross entropy loss. We trained for 60000 steps with batch size 10. We used the Adam optimizer [13] with a start learning of `1e-2` and a reduction of the learning rate by 70% every 15000 steps. Training took up to 2 days on a system with 4 CPU cores, 30 GB of RAM, and an NVIDIA GeForce GTX 1080 Ti GPU.

The input to the Tensorfield network and the Se(3) Transformer are relative x-y-z positions of each point w.r.t. their neighbours. To guarantee equivariance, these inputs are provided as fields of degree 1. For the '+z' versions, however, we deliberately break the SE(3) equivariance by providing additional and relative z-position as two additional scalar fields (i.e. degree 0), as well as relative x-y positions as a degree 1 field (where the z-component is set to 0).

### D.1.2 Number of Input Points

Limiting the input to 128/200 points in our experiments on ScanObjectNN was not primarily due to computational limitations: we conducted experiments with up to 2048 points, but without performance improvements. We suspect this is due to the global pooling. Examining cascaded pooling via attention is a future research direction. Interestingly, when limiting other methods to using 128 points, the SE(3)-Transformer outperforms the baselines (PointCNN: $80.3 \pm 0.8\%$, PointGLR: $81.5 \pm 1.0\%$, DGCNN: $82.2 \pm 0.8\%$, **ours**: $85.0 \pm 0.7\%$). It is worth noting that these methods were explicitly designed for the well-studied task of point classification whereas the SE(3)-Transformer was applied as is. Combining different elements from the current state-of-the-art methods with the geometrical inductive bias of the SE(3)-Transformer could potentially yield additional performance gains, especially with respect to leveraging inputs with more points. It is also worth noting that the SE(3)-Transformer was remarkably stable with respect to lowering the number of input points in an ablation study (16 points: $79.2\%$, 32 points: $81.4\%$, 64 points: $82.5\%$, 128 points: $85.0\%$, 256 points: $82.6\%$).

### D.1.3 Sample Complexity

Equivariance is known to often lead to smaller sample complexity, meaning that less training data is needed (Fig. 10 in Worrall et al. [40], Fig. 4 in Winkels and Cohen [38], Fig. 4 in Weiler et al. [37]). We conducted experiments with different amounts of training samples $N_{samples}$ from the ScanObjectNN dataset. The results showed that for all $N_{samples}$, the SE(3)-Transformer outperformed the Set Transformer, a non-equivariant network based on attention. The performance delta was also slightly higher for the smallest $N_{samples}$ we tested ($3.1\%$ of the samples available in the training split of ScanObjectNN) than when using all the data indicating that the SE(3)-Transformer performs particularly well on small amounts of training data. However, performance differences can be due to multiple reasons. Especially for small datasets, such as ScanObjectNN, where the performance does not saturate with respect to amount of data available, it is difficult to draw conclusions about sample complexity of one model versus another. In summary, we found that our experimental results are in line with the claim that equivariance decreases sample complexity in this specific case but do not give definitive support.

### D.1.4 Baselines

**DeepSet Baseline** We originally replicated the implementation proposed in [46] for their object classification experiment on ModelNet40 [41]. However, most likely due to the relatively small number of objects in the ScanObjectNN dataset, we found that reducing the model size helped the performance significantly. The reported model had 128 units per hidden layer (instead of 256) and no dropout but the same number of layers and type of non-linearity as in [46].

**Set Transformer Baseline** We used the same architecture as [16] in their object classification experiment on ModelNet40 [41] with an ISAB (induced set attention block)-based encoder followed by PMA (pooling by multihead attention) and an MLP.

## D.2 Relational Inference

Following Kipf et al. [14], we simulated trajectories for 5 charged, interacting particles. Instead of a 2d simulation setup, we considered a 3d setup. Positive and negative charges were drawn as Bernoulli trials ($p = 0.5$). We used the provided code base `https://github.com/ethanfetaya/nri` with the following modifications: While we randomly sampled initial positions inside a $[-5, 5]^3$ box, we removed the bounding-boxes during the simulation. We generated 5k simulation samples for training and 1k for testing. Instead of phrasing it as a time-series task, we posed it as a regression task: The input data is positions and velocities at a random time step as well as the signs of the charges. The labels (which the algorithm is learning to predict) are the positions and velocities 500 simulation time steps into the future.

**Training Details**  We trained each model for 100,000 steps with batch size 128 using an Adam optimizer [13]. We used a fixed learning rate throughout training and conducted a separate hyper parameter search for each model to find a suitable learning rate.

**SE(3)-Transformer Architecture**  We trained an SE(3)-Transformer with 4 equivariant layers, where the hidden layers had representation degrees $\{0, 1, 2, 3\}$ and 3 channels per degree. The input is handled as two type-1 fields (for positions and velocities) and one type-0 field (for charges). The learning rate was set to `3e-3`. Each layer included attentive self-interaction.

We used 1 head per attention mechanism yielding one attention weight for each pair of points but across all channels and degrees (for an implementation of multi-head attention, see Appendix D.3). For the query embedding, we used the identity matrix. For the key embedding, we used a square equivariant matrix preserving the number of degrees and channels per degree.

**Baseline Architectures**  All our baselines fulfill permutation invariance (ordering of input points), but only the Tensor Field network and the linear baseline are SE(3) equivariant. For the **Tensor Field Network**[28] baseline, we used the same hyper parameters as for the SE(3) Transformer but with a linear self-interaction and an additional norm-based nonlinearity in each layer as in Thomas et al. [28]. For the **DeepSet**[46] baseline, we used 3 fully connected layers, a pooling layer, and two more fully connected layers with 64 units each. All fully connected layers act pointwise. The pooling layer uses max pooling to aggregate information from all points, but concatenates this with a skip connection for each point. Each hidden layer was followed by a LeakyReLU. The learning rate was set to `1e-3`. For the **Set Transformer**[16], we used 4 self-attention blocks with 64 hidden units and 4 heads each. For each point this was then followed by a fully connected layer (64 units), a LeakyReLU and another fully connected layer. The learning rate was set to `3e-4`.

For the **linear baseline**, we simply propagated the particles linearly according to the simulation hyperparamaters. The linear baseline can be seen as removing the interactions between particles from the prediction. Any performance improvement beyond the linear baseline can therefore be interpreted as an indication for relational reasoning being performed.

## D.3 QM9

The QM9 regression dataset [21] is a publicly available chemical property prediction task consisting of 134k small drug-like organic molecules with up to 29 atoms per molecule. There are 5 atomic species (Hydrogen, Carbon, Oxygen, Nitrogen, and Flourine) in a molecular graph connected by chemical bonds of 4 types (single, double, triple, and aromatic bonds). 'Positions' of each atom, measured in ångtröms, are provided. We used the exact same train/validation/test splits as Anderson et al. [1] of sizes 100k/18k/13k.

The architecture we used is shown in Table 4. It consists of 7 multihead attention layers interspersed with norm nonlinearities, followed by a TFN layer, max pooling, and two linear layers separated by a ReLU. For each attention layer, shown in Fig. 7, we embed the input to half the number of feature channels before applying multiheaded attention [31]. Multiheaded attention is a variation of attention, where we partition the queries, keys, and values into $H$ *attention heads*. So if our embeddings have dimensionality $(4, 16)$ (denoting 4 feature types with 16 channels each) and we use $H = 8$ attention heads, then we partition the embeddings to shape $(4, 2)$. We then combine each of the 8 sets of shape $(4, 2)$ queries, keys, and values individually and then concatenate the results into a single vector of the original shape $(4, 16)$. The keys and queries are edge embeddings, and thus the embedding matrices are of TFN type (c.f. Eq. (8)). For TFN type layers, the radial functions are learnable maps. For these we used neural networks with architecture shown in Table 5.

For the norm nonlinearities [40], we use

$$\texttt{Norm ReLU}(\mathbf{f}^\ell) = \texttt{ReLU}(\texttt{LN}\left(\|\mathbf{f}^\ell\|\right)) \cdot \frac{\mathbf{f}^\ell}{\|\mathbf{f}^\ell\|}, \qquad \text{where } \|\mathbf{f}^\ell\| = \sqrt{\sum_{m=-\ell}^{\ell} (f_m^\ell)^2}, \qquad (52)$$

where $\texttt{LN}$ is layer norm [2] applied across all features within a feature type. For the TFN baseline, we used the exact same architecture but we replaced each of the multiheaded attention blocks with a TFN layer with the same output shape.

The input to the network is a sparse molecular graph, with edges represented by the molecular bonds. The node embeedings are a 6 dimensional vector composed of a 5 dimensional one-hot embedding of the 5 atomic species and a 1 dimension integer node embedding for number of protons per atom. The edges embeddings are a 5 dimensional vector consisting of a 4 dimensional one-hot embedding of bond type and a positive scalar for the Euclidean distance between the two atoms at the ends of the bond. For each regression target, we normalised the values by mean and dividing by the standard deviation of the training set.

We trained for 50 epochs using Adam [13] at initial learning rate $\texttt{1e-3}$ and a single-cycle cosine rate-decay to learning rate $\texttt{1e-4}$. The batch size was 32, but for the TFN baseline we used batch size 16, to fit the model in memory. We show results on the 6 regression tasks not requiring thermochemical energy subtraction in Table 3. As is common practice, we optimised architectures and hyperparameters on $\varepsilon_{\text{HOMO}}$ and retrained each network on the other tasks. Training took about 2.5 days on an NVIDIA GeForce GTX 1080 Ti GPU with 4 CPU cores and 15 GB of RAM.

Table 4: QM9 Network architecture: $d_{\text{out}}$ is the number of feature types of degrees $0, 1, ..., d_{\text{out}} - 1$ at the output of the corresponding layer and $C$ is the number of multiplicities/channels per feature type. For the norm nonlinearity we use ReLUs, preceded by equivariant layer norm [37] with learnable affine transform.

| No. repeats | Layer Type | $d_{\text{out}}$ | $C$ |
|---|---|---|---|
| 1x | Input | 1 | 6 |
| 1x | Attention: 8 heads | 4 | 16 |
|  | Norm Nonlinearity | 4 | 16 |
| 6x | Attention: 8 heads | 4 | 16 |
|  | Norm Nonlinearity | 4 | 16 |
| 1x | TFN layer | 1 | 128 |
| 1x | Max pool | 1 | 128 |
| 1x | Linear | 1 | 128 |
| 1x | ReLU | 1 | 128 |
| 1x | Linear | 1 | 1 |

Table 5: QM9 Radial Function Architecture. $C$ is the number of output channels at each layer. Layer norm [2] is computed per pair of input and output feature types, which have $C_{\text{in}}$ and $C_{\text{out}}$ channels each.

| Layer Type | $C$ |
|---|---|
| Input | 6 |
| Linear | 32 |
| Layer Norm | 32 |
| ReLU | 32 |
| Linear | 32 |
| Layer Norm | 32 |
| ReLU | 32 |
| Linear | $d_{\text{out}} * C_{\text{in}} * C_{\text{out}}$ |

Figure 7: Attention block for the QM9 dataset. Each component is listed with a tuple of numbers representing the output feature types and multiplicities, so $(4, 32)$ means feature types $0, 1, 2, 3$ (with dimensionalities $1, 3, 5, 7$), with 32 channels per type.

## D.4 General Remark

Across experiments on different datasets with the SE(3)-Transformer, we made the observation that the number of representation degrees have a significant but saturating impact on performance. A big improvement was observed when switching from degrees $\{0, 1\}$ to $\{0, 1, 2\}$. Adding type-3 latent representations gave small improvements, further representation degrees did not change the performance of the model. However, higher representation degrees have a significant impact on memory usage and computation time. We therefore recommend representation degrees up to 2, when computation time and memory usage is a concern, and 3 otherwise.

## Footnotes

[5]Over a field of characteristic zero.