[Reviews · NeurIPS 2020]

Review 1

Summary and Contributions: The authors propose how to extend Transformer Architectures to be equivariant under rigid motions.

Strengths: The paper is clearly written. Transformers are a very successful architecture, so having a way to extend them to be equivariant under rigid motions is clearly valuable and significant. This hasn't been done before as far as I can tell.

Weaknesses: The novelty relative to Tensor Field Networks seems quite limited to me, it's a quite straightforward way to combine them with transformers. The experimental results are not amazing, i.e. it's not clearly better than the corresponding baselines. The problems considered are somewhat toy. Many of their results don't have any indication of variance in their results, i.e. error bars. This is especially important for small regression problems like the QM9 dataset, for these kind of problems in my experience neural networks error bars can get quite large.

Correctness: The main claim that they built an architecture that is equivariant under rigid motions most certainly seems correct to me.

Clarity: The paper is well written and easy to follow, I'm speaking as someone who is very familiar with group theory though.

Relation to Prior Work: The discussion of related work in the paper is quite deeply intertwined with the discussion of generic mathematical background, this makes paper read more nicely, but makes the distinction from previous work less pronounced. But overall I think the authors give a fair account of related work.

Reproducibility: Yes

Additional Feedback:


Review 2

Summary and Contributions: The paper proposes to add attention layers to the Tensor Field Networks (TFN) [1], which increases the model expressivity while preserving SE(3) equivariance. Secondary contributions are a more efficient TFN implementation, and a modified self-interaction layer with attention. The model is shown to outperform an attention-only and a TFN baseline on a few tasks. [1] Thomas et al, "Tensor Field Networks: Rotation- and Translation-Equivariant Neural Networks for 3D Point Clouds", 2018.

Strengths: The approach improves upon previous methods by mitigating the loss of representational power when constraining the filters to achieve equivariance. The theory is sound and the implementation seems efficient and likely to be useful for future work, if the code is released.

Weaknesses: 1) The major weakness is that the paper can be seen as a simple extension of the Tensor Field Networks (TFN) [1] by adding attention layers, so novelty is somewhat limited. I do realize that some adaptations are necessary, e.g., using TFN layers to produce query/keys/values and showing that the attention weights are invariant, so there is value in the contributions. 2) I'd like to see more discussion on limitations, specifically the computational cost. It's my understanding that the original TFN [1] was impractical for the point clouds sizes typically used for ModelNet and ShapeNet (a few thousand points). The proposed method includes a faster spherical harmonics implementation; is that part the bottleneck of the TFN? How big is the speed-up? Some numerical comparison of training/inference time of different methods would be enlightening. My impression is that the proposed method is still much slower than the PointNet-based alternatives (including some that are rotation-invariant; see next point). The proposed model seems to use 10-20x fewer points, a dataset about 10x smaller than ShapeNet and still takes two days to train. 3) There are several recent works on rotation-invariant point cloud analysis that were not discussed [2, 3, 4]; while the proposed approach is more theoretically sound and can potentially result in more expressive models, I think some discussion of current work and the gap between theory and practice is warranted. These other methods often show results for rotated ModelNet40 classification, part segmentation, and ShapeNet retrieval (SHREC'17); evaluation on these datasets would strengthen the submission. [1] Thomas et al, "Tensor Field Networks: Rotation- and Translation-Equivariant Neural Networks for 3D Point Clouds", 2018. [2] Zhang et al, "Rotation Invariant Convolutions for 3D Point Clouds Deep Learning", 3DV'19. [3] Rao et al, "Spherical Fractal Convolutional Neural Networks for Point Cloud Recognition", CVPR'19. [4] Chen et al, "ClusterNet: Deep hierarchical cluster network with rigorously rotation-invariant representation for point cloud analysis", CVPR'19.

Correctness: Yes.

Clarity: Yes.

Relation to Prior Work: Somewhat; I believe recent rotation invariant point cloud models should be discussed (see my point 3 in weaknesses).

Reproducibility: Yes

Additional Feedback: - typo in Fig 1 caption: "euqivariance" - L553 (appendix): what is a quadratic equivariant matrix? === POST-REBUTTAL UPDATE === After reading the reviews and discussion I think all reviewers agree about the strengths and weaknesses of the paper. The approach might be considered incremental with respect to TFN and Set Transformers, and the experiments are not too strong. On the other hand, the method is theoretically sound and clearly superior to both the TFN and Set Transformers, and the code to be released will probably be useful for future research; in particular the fast spherical harmonics library may have broader applicability. Overall I think the merits outweigh the flaws.


Review 3

Summary and Contributions: This submission introduces the SE(3)-Transformers: a new variant of the self-attention layer which is equivariant to roto-translations, i.e. transformations living in the SE(3) space, aimed at processing 3D data. An attention module consists of two main components: an embedding of the input data (value embedding) and the attention weights describing how much each of the input data attend to each other. The proposed method uses Tensor Field Networks [23] to obtain an equivariant value embedding. It extends the convolution layer of TFNs to include attention weights and values. It further shows that when TFNs are used to compute also the query and key matrices in an equivariant way, the resulting attention weights are invariant wrt the input pose. Equivariant input embeddings and invariant attention weights result in an overall equivariant self-attention layer. To realize attention efficiently, a graph for each 3D point and the nearest neighbors is built so that features can flow along the edges only in a local area and the total complexity is linear rather than quadratic. In order to validate the performance of the method, the authors perform experiments on three different tasks: predictions of locations and velocities of the particles, 3D real-world object classification task on the standard ScanObjectNN dataset, and molecular regression. In the last two cases, SE(3)-Transformers achieve good results, but do not improve the previous state of the art.

Strengths: 2.1 - The processing of point clouds with deep learning algorithms is challenging due to several factors, one of the most important one being invariance to the pose of the input data. The presented work addresses this problem in an interesting way introducing SE(3)-equivariant attention layers. Transformers have been disruptive in NLP, and its interesting to study if they can bring similar breakthroughs in other applications. Hence, I liked the idea of developing a roto-translation equivariant attention layers based on them. 2.2 - Novelty. Even though SE(3)-Transformers results mainly from the smart combination of two pre-existing works, i.e. Set-Transformers [12] and the SE(3)-equivariant TFN neural networks [23], such combination is not trivial. 2.3 - The work is relevant for the NeurIPS community, as it addresses an important open issue like equivariant processing of 3D data by means of a method which has delivered outstanding performance in other fields. Moreover, it provides an important theoretical contribution for processing of point clouds with deep learning algorithms. 2.4 - The work is presented in a clear and convincing way together with a strong theoretical background accompanied by the related demonstrations in the supp. material. 2.5 - Equivariance and reasonable effectiveness of the propose method is empirically validated in the experimental results on different applications.

Weaknesses: 3.1 Experimental results. In the two most challenging benchmarks (ScanObjectNN and QM9) the proposed Transformer achieves performance inferior to the best performing methods. In particular, in the ScanObjectNN experiment, the authors had to break the SE(3) equivariance of the network by providing the z coordinate as additional input feature, to achieve satisfactory performance. From the chart (a) of figure 4, it looks like the performance of the fully equivariant Transformer is around 73%, instead of 85%. The authors should add this value to table 2. Overall, these results are disappointing, in particular in light of the higher complexity of the model compared to competitors (e.g. to PointNet++ which achieves 84% on ScanObjectNN), which does not deliver a boost in performance. This could also be due to the selected benchmark, as noted by the authors, but the same relative rank with respect to state-of-the-art methods repeats also on the QM9 dataset. 3.2 Practical applicability, in particular scalability with respect to the number of input points is a weakness. A very small number of points are used in the experimental results to process point clouds (eg. 128 instead of the original 2048). This value may not be adequate when working with data covering very large areas such as indoor and outdoor scenes and limits the applicability of the proposed methods to tasks which can be solved effectively only by using hundreds of points.

Correctness: Overall, the evaluation presented in the main paper is conducted in a fair way comparing against state-of-the-art method and with proper metrics. Moreover, in the supplementary materials the authors provide all the details adopted for the experiments. However, in the introduction the authors claims 3 main contributions, two of which are how SE(3)-Transformer resolves the angularly constrained filters problem of SE(3)-equivariant neural networks and a new implementation of spherical harmonics which is faster than the previous ones. Although these two claims are theoretically well motivated in section 3.2 of the main paper and section C of supplementary material respectively, ablation studies or comparisons would strengthen the claims. Similarly, the authors claim their method can operate "on large point clouds with varying number of points" (lines 7-8). As they use a fixed input size of 128 points, they should tone down this and similar claims in the paper (e.g. lines 268 in the conclusions).

Clarity: The main paper is easy to follow and the single steps to decompose the method are clearly explained in section 3. Further details and exhaustive demonstrations about the mathematical model behind the SE(3)-Transformers are given in the supplementary and enrich the submission. Minor points: - In equation (13), what does f^l_{in,i,c'} represent, assuming c' are the output channels as in (12)? - It is not clear how input features are computed for 3D point cloud. Please add this detail to the main text.

Relation to Prior Work: This work is built by combining ideas from Tensor Field Networks and Set Transformer. Indeed, Tensor Field Networks deploy an SE(3) equivariant neural network and Set Transformer apply self-attention algorithm to point clouds. Their basic notions are well explained in the background section, while in the experimental results the authors demonstrate that by merging these concepts they can outperform Set Transformer and Tensorf Field Network with or without data augmentation in 3D Object classification. However, the difference wrt these work should be highlighted in the introduction. Relation toSpherical CNNs and its variants, e.g. PRIN, another approach to equivariant/invariant processing of 3D data, should be discussed in the Background section. Taco S. Cohen and Max Welling. Spherical Cnns. International Conference on Learning Representations (ICLR), 2018. Yang You et al., Pointwise Rotation-Invariant Network with Adaptive Sampling and 3D Spherical Voxel Convolution, AAAI 2020

Reproducibility: Yes

Additional Feedback: * Line 6: "less training data" -> i think this statement is misleading. It would be better to specify that no data augmentation is needed to learn invariance to rotation. * Line 8: "operate on large point clouds with varying number of points" -> the data adopted in the experimental session capture object and seems not to be so "large". Moreover, the method uses a fixed number of points, (e.g. 128) so is not clear how it can handle varying point cloud density. * Caption of Figure 1: "euqivariance" -> "equivariance" * Line 41: "symmetries of the task" -> This sentence is often repeated in the paper, but I find it unclear. In my opinion, symmetries are due to the data and not to the specific task. I believe that a longer explanation of this concept may improve the clarity of the work. * Line 182: "and input representation degrees k" -> maybe a typo? * In the introduction, the authors should put more emphasis on the difference between the proposed method and Set-Transformers and Tensor Field Network. * Table 2: The results without the Z coordinate as input (SE(3)-Transformer), should be added. * Table 3: Bold is not used for any method. ==== POST-REBUTTAL UPDATE ===== Having read the rebuttal and the other reviews. I decided to retain my score. In particular, I'm positive about the novelty of the paper: I agree that all the ingredients were already there as pointed out by other reviewers, but I still consider their combination, given their complexity, not obvious and worth to be shared with the community. On the other hand, the comments in the rebuttal about the experimental results are not convincing: it is strange that using more than 128 points did not increase performance for the proposed method, as 128 points is a very coarse resolution. Is this hinting to some limited capabilities of SE(3)-Transformers to capture detailed geometric structures? Could this be due to the need to create a local graph, and selection of the right hyperparameters (i.e. number of neighbors) is harder to generalize across models when 3D clouds have finer details? I don't see how global pooling, as speculated by the authors, could be a valid explanation. Moreover, the performance on ScanObjectNN without the additional z coordinate as input, which breaks equivariance, are really mediocre. The authors state that this does not show a weakness of the approach, but I don't agree: the fact that the fully SE(3) equivariant network cannot learn to handle a non-fully-rotated dataset seems indeed a limitation, as the other approaches reported in the table do not need any additional hint to achieve high performance on the same dataset. On the positive side, the author's response better clarified how remarkable the speed-up of their implementation over the standard TFN implementation is. Probably the authors should emphasize this more in the main paper, beyond the introduction, e.g. in the experimental results.


Review 4

Summary and Contributions: This paper introduces SE(3)-Transformer, a new model that combines self-attention and equivariance to SE(3) which shows improved performance on a variety of 3D point cloud tasks. The proposed model is provable equivariant to SE(3) transformations, which is an impactful and desirable property for pt cloud models.

Strengths: In addition to the contributions above, this paper is generally well-written, the theory is self-contained which improves readability. The experimental details are all included for reproducibility.

Weaknesses: *************************** Final comments: Both the novelty of the idea and significance of the experimental results are incremental. The proposed approach is a straightforward combination of existing pieces, and while the results show improvements over TFN (expected) it is difficult to draw any broader conclusions from the experiments (see the comments from the other reviews). Nonetheless, this is a complete, correct, and well-written paper that is relevant to the NeurIPS community. The authors could do a better job of highlighting the motivations/contributions as provided in the author response (e.g. the importance of being able to generate Spherical Harmonics online should be emphasized, and this would highlight the impact of any provided code). *************************** On first reading it is hard to understand the exact interplay between the two primary themes of this paper: (a) SE(3) equivariance and (b) self-attention. In other words it is not obvious how (b) depends on (a) or vice versa. From eq 10 through eq 11, it seems like nothing needs to be done to maintain attention weight invariance to SE(3) transformations. It is not clear if the proposed method is as straightfoward as adding a self-attention step to Tensor Field Networks or if there is some underlying complexity that needed consideration. If I understand the method correctly, one of the limiting factors of this approach would be related to computational performance/complexity/memory. It would be helpful if a discussion could be presented in the main paper since it seems like this might be a major issue with scalability. Related to this is the design decision for representation degrees in the model. There is a very short discussion in D.4 (supplemental) but this dimension is worth discussing further. ScanObjectNN: Table 2 could be a bit misleading. Comparing performance of proposed method at 128pts against baselines at 1024 pts implies the baselines *need* 1024 points and will fail at 128, or that proposed method would be much faster than others using 1024 pts. Such conclusions would need to be supported with quantitative evidence.

Correctness: Method seems correct, did not check all details.

Clarity: Paper is well written aside from a few points of confusion (see above).

Relation to Prior Work: Related work review is clear.

Reproducibility: Yes

Additional Feedback: Based on the high level comments about the contributions and limitations of the proposed method, my current stance is borderline, but am willing to raise the score if clarity to those issues is provided. N-body simulations: Were the non-equivariant models trained with augmentation (same trajectories observed under different transformations), or is each training sample a new simulation? L109: "set Q=I, i=l". It is unclear what "i=l" means here and it reads correct without it.

[Author Response · NeurIPS 2020]

1. We thank the reviewers for their in-depth and constructive reviews. We are happy that idea, presentation, implementation
2. and experimental validation are well perceived and that all reviewers stated that they lean towards acceptance.

3. **Novelty with respect to Tensor Field Networks (TFN)** The reviewers state that our work is novel (**R3**), impactful
4. (**R4**), valuable (**R1**), sound (**R2**), and an important theoretical contribution (**R3**). **R1**, **R2**, & **R4** state that the approach
5. could be seen as a straightforward extension of TFNs; however, as **R3** states, such extension is nontrivial. We are
6. able to derive invariant weights only through making keys and queries dependent on relative positions. This differs
7. significantly from regular attention and is not an obvious choice. This extends TFNs to the graph setting, with the bonus
8. of permitting edge features. Furthermore, this is one of the first examples of a nonlinear equivariant layer. In Section
9. 3.2, we show our proposed approach relieves the strong angular constraints on the filter compared to TFNs, therefore
10. adding representational capacity. This constraint has been pointed out by several authors in the equivariance literature
11. to limit performance severely (Weiler & Cesa, 2019). We will emphasize this comparison in the introduction (**R3**).

12. **Computational Cost / Scalability** Concerning scalability (**R2**, **R3**, **R4**), wrt the original TFN implementation (**R2**),
13. we will extend the discussion. The spherical harmonics (SH) have to be computed on the fly for point cloud methods, a
14. bottleneck of TFNs. The TFN authors ameliorated this by restricting the maximum type of feature to type-2, trading
15. expressivity for speed. We built a faster SH library (10x faster on CPU, 100-1000x on GPU than existing libraries) and
16. can handle any SH type. We will include a speed comparison. E.g., for a ScanObjectNN model, we achieve $\sim 22\times$
17. speed up of the forward pass compared to a network built with SH from the `lielearn` library. **We will release code**
18. for the camera-ready (**R2**). In the meantime, a recipe for the GPU-based SH generation is already in Appendix C.

19. (**R3**): (1) We conducted experiments with up to 2048 points, but with no significant performance improvements. We
20. suspect this is due to the global pooling. Examining cascaded pooling via attention is a future research direction. (2) For
21. the QM9 dataset, we efficiently deal with varying point cloud sizes, leveraging the DGL library for GPU parallelisation.

22. **Experimental Results** The experiments validate effectiveness and equivariance of the approach (**R3**), showing that the
23. SE(3) Transformer consistently outperforms non-equivariant self-attention and TFN (**R2**) while providing reproducabil-
24. ity (**R1**,**R2**,**R3**,**R4**). It is also noted we do not improve on the previous SOTA on ScanObjectNN and QM9. It is worth
25. noting the TFN baseline we report is significantly scaled up compared to the original implementation (more channels,
26. higher degrees & more layers), enabled by the efficiency improvements described above. The performance difference
27. between TFN & SE(3) Transformer comes on top of those improvements. We want to stress this is the first time an
28. equivariant point cloud network based on irreps reports competitive results on object classification. While these results
29. are not SOTA, we are close. Furthermore, on ScanObjectNN, the baselines are specifically designed for that task, and
30. we introduce a component rather than an architecture, so could in theory combine the benefits of the SE(3) Transformer
31. with, say, PointNet++. From the point of view of the equivariance literature, we have some work to do to catch up with
32. non-equivariant works, but we feel the SE(3) Transformer makes a meaningful step forward in closing that gap. This
33. work brings equivariant methods closer to being a useful tool in the practitioner's toolbox. As correctly noted by **R3**,
34. we deliberately break SE(3) equivariance for ScanObjectNN. Feeding the z-coord. as a separate, scalar input makes the
35. network SE(2) equivariant. In our opinion, this does not indicate a weakness of the approach of SE(3) invariance in
36. general. Instead it shows object classification datasets are not fully symmetric. We happily include the SE(3) equivariant
37. version in table 2. Why not ModelNet40 (**R2**)? Due to an internal policy, we were limited to datasets with proper
38. licensing, which ModelNet40 does not provide. ScanObjectNN is a more recent dataset providing a tougher alternative
39. to ModelNet40 based on noisy real world sensor data. **Do the baselines *need* 1024 points (R4)?** Initial experiments
40. show, when training & testing on 128 points only, we do outperform the baselines (PointCNN: $80.3 \pm 0.8\%$, PointGLR:
41. $81.5 \pm 1.0\%$, DGCNN: $82.2 \pm 0.8\%$, **ours**: $85.0 \pm 0.7\%$). A more detailed analysis will be added to the appendix.

42. On QM9, the only network which beats the SE(3) Transformer on all tasks is LieConv, a concurrently developed
43. approach published after the submission of this work. The equivariance of LieConv networks is based on (left-) regular
44. representations, coming with its own set of dis/advantages (e.g. stochastic forward pass and complicated extension to
45. higher degree input representations). Cormorant is on par but uses expensive Clebsch-Gordan transform nonlinearity -
46. the authors state that training is unstable, an issue we do not have. (**R1**) Error bars: Table 1 contains error bars, in table
47. 2 it is in the caption. In Table 3 we shall add them. For QM9 stddev is small - e.g. we got $\sim 1$ meV for the $\varepsilon_{\text{HOMO}}$ task.

48. **Other Questions** (**R2**) L553: quadratic - should say 'square'; (**R2**,**R3**) **Related Work** - We thank the reviewers for
49. the pointers and will discuss these references in the paper; (**R3**) L6 - the claim of **improved sample complexity** is
50. supported by [Fig. 10 Worrall et al. (2017), Fig. 4 Winkels & Cohen (2018), Fig. 2 Bekkers et al. (2018), Fig. 4 Weiler
51. et al., (2018)], we will include a quantitative analysis; (**R3**) We will add bold for Table 3; (**R3**) Clarify 'symmetries
52. of the task': we use the definition of symmetry from Mallat, (2016) "Understanding Deep Convolutional Networks"
53. where symmetry is a property of a function/task; (**R3**) How are input features computed for 3D point cloud? We use
54. relative coordinates as features for the first layer (see appendix D.1); (**R3**) Equation (13) - $c'$ and $c$ are placeholders for
55. input channels; (**R4**) We did not augment data for the n-body experiment, but the data is sampled uniformly across all
56. orientations; (**R4**) L109 - '$i = l$' should be deleted. Finally, we thank the reviewers for all mentioned typos.

[Meta-Review · NeurIPS 2020]

All reviewers recommend acceptance. The main concern, shared by all reviewers, is the limited novelty of the work relative to Tensor Field Networks. Other concerns included some missing related work, and some issues with the empirical evaluation. Although the discussion did not really change anyone's view of this work in the end, the fact that it was solidly executed plays in its favour. While the combination of ideas from transformers / self-attention and tensor field networks is a logical next step following on from the latter work, I concur with R3 that the concretisation of this idea is not necessarily obvious and worth sharing with the community. I would like to join the reviewers in recommending that the authors try to emphasize the potential impact of the ability to generate spherical harmonics online in the revised version of their manuscript.